# A phosphorylation of RIPK3 kinase initiates an intracellular apoptotic pathway that promotes prostaglandin$_{2\alpha}$-induced corpus luteum regression

Dianrong Li[1,2], Jie Chen[1,2], Jia Guo[1,2], Lin Li[1,2], Gaihong Cai[1,2], She Chen[1,2], Jia Huang[3], Hui Yang[3], Yinhua Zhuang[1,2], Fengchao Wang[1,2], Xiaodong Wang[1,2]*

[1]National Institute of Biological Sciences, Beijing, China; [2]Tsinghua Institute of Multidisciplinary Biomedical Research, Tsinghua University, Beijing, China; [3]Institute of Neuroscience, State Key Laboratory of Neuroscience, Key Laboratory of Primate Neurobiology, CAS Center for Excellence in Brain Science and Intelligence Technology, Shanghai Research Center for Brain Science and Brain-Inspired Intelligence, Shanghai Institutes for Biological Sciences, Chinese Academy of Sciences, Shanghai, China

**Abstract** Receptor-interacting serine/threonine-protein kinase 3 (RIPK3) normally signals to necroptosis by phosphorylating MLKL. We report here that when the cellular RIPK3 chaperone Hsp90/CDC37 level is low, RIPK3 also signals to apoptosis. The apoptotic function of RIPK3 requires phosphorylation of the serine 165/threonine 166 sites on its kinase activation loop, resulting in inactivation of RIPK3 kinase activity while gaining the ability to recruit RIPK1, FADD, and caspase-8 to form a cytosolic caspase-activating complex, thereby triggering apoptosis. We found that PGF$_{2\alpha}$ induces RIPK3 expression in luteal granulosa cells in the ovary to cause luteal regression through this RIPK3-mediated apoptosis pathway. Mice carrying homozygous phosphorylation-resistant RIPK3 S165A/T166A knockin mutations failed to respond to PGF$_{2\alpha}$ but retained pro-necroptotic function, whereas mice with phospho-mimicking S165D/T166E homozygous knock-in mutation underwent spontaneous apoptosis in multiple RIPK3-expressing tissues and died shortly after birth. Thus, RIPK3 signals to either necroptosis or apoptosis depending on its serine 165/threonine 166 phosphorylation status.

*For correspondence:
wangxiaodong@nibs.ac.cn

Competing interests: The authors declare that no competing interests exist.

## Introduction

Regulated cell death with distinctive associated morphological changes happens either in the form of apoptosis or necrosis (*Elmore, 2007*; *Fink and Cookson, 2005*; *Kroemer et al., 2009*; *Wallach et al., 2016*). Apoptosis is executed by intracellular caspases, particularly caspase-3, whose cleavage of several intracellular substrates gives rise to characteristic features of apoptotic cell death. Caspase-3 is activated by upstream caspase-8 or caspase-9, which are known to be activated respectively by being part of plasma membrane-associated death-inducing signaling complexes induced by the TNF family of cytokines, or being part of the cytosolic apoptosome, which is formed as a result of cytochrome c release from mitochondria (*Elmore, 2007*; *Li et al., 1997*; *Wang et al., 2008*). Cells that die by apoptosis maintain their plasma membrane integrity and break down into small vesicles phagocytosed by macrophages (*Elmore, 2007*; *Kroemer et al., 2009*). Cells that die by regulated necrosis are executed by membrane-disrupting proteins that, when activated, translocate from the cytosol to the plasma membrane and disperse cellular contents into the cell

surroundings. This process is often associated with inflammatory response (*Christofferson and Yuan, 2010*; *Vandenabeele et al., 2010*; *Wallach et al., 2016*).

One form of regulated necrosis, necroptosis, is executed by a pseudokinase, namely mixed lineage kinase-domain protein, MLKL (*Sun et al., 2012*). MLKL is a substrate of receptor-interacting kinase RIPK3, which becomes activated either by (1) the TNF family of cytokines through a related kinase RIPK1 that interacts with RIPK3 through a RIP homotypic interaction motif (RHIM) domain or (2) by other RHIM domain-containing proteins, such as TRIF or ZBP1/DAI, that relay necroptotic signals from Toll-like receptors or intracellular Z-RNA, respectively (*Cho et al., 2009*; *Degterev et al., 2008*; *He et al., 2009*; *Sun et al., 2012*; *Upton et al., 2012*; *Zhang et al., 2009*; *Zhang et al., 2020*). Active RIPK3 phosphorylates MLKL at serine 357/threonine 358 (human origin) sites to induce oligomerization and translocation from the cytosol to the plasma membrane and cause loss of membrane integrity (*Cai et al., 2014*; *Chen et al., 2014*; *Sun et al., 2012*; *Wang et al., 2014*).

Surprisingly, a kinase-dead mutant RIPK3, D161N, spontaneously causes host cells to undergo apoptosis, resulting in embryonic lethality in mice bearing such homozygous mutation (*Newton et al., 2014*). On the other hand, mice homozygous with another kinase-dead mutant, K51A RIPK3, do not share this phenotype. However, cells that harbor this mutation are sensitive to apoptosis induction when treated with several small-molecule RIPK3 inhibitors that bind to its ATP-binding pocket (*Mandal et al., 2014*). Whether this apoptosis-inducing activity of RIPK3 is an anomaly of the mutant protein, the small-molecule inhibitor bound form, or a true physiological function of RIPK3 remains to be resolved.

Apoptosis and necroptosis play important physiological and pathological roles in mammals (*Christofferson and Yuan, 2010*; *Elmore, 2007*; *Vandenabeele et al., 2010*; *Wallach et al., 2016*). Apoptosis has essential functions during animal development and maintaining cell homeostasis in adults (*Elmore, 2007*; *Fink and Cookson, 2005*). Necroptosis functions include antiviral infections, tissue damage response, and male reproductive system aging (*Cho et al., 2009*; *He and Wang, 2018*; *Li et al., 2017*).

One of the important physiological functions of apoptosis in adult mammals is the development and function of the ovary (*Perez et al., 1997*; *Rimon-Dahari et al., 2016*; *Rolaki et al., 2005*; *Tilly, 2001*). From the neonatal stage to sex maturation, most eggs within primordial follicles fail to develop and undergo a process called atresia – a process driven by apoptosis of the egg and surrounding granulosa cells (*Rimon-Dahari et al., 2016*; *Tilly and Sinclair, 2013*). After sex maturation, follicles develop from primordial to primary and secondary follicles accompanied by the proliferation of surrounding granulosa and theca cells (*Rimon-Dahari et al., 2016*; *Sánchez and Smitz, 2012*; *Tilly and Sinclair, 2013*). After egg release from the follicle during ovulation, the remaining granulosa and theca cells become progesterone-producing sources, namely corpus luteum, which is easily recognizable under a microscope (*Rolaki et al., 2005*). If the egg is unfertilized, the corpus luteum dies through apoptosis and becomes corpus albicans, a scar tissue with a mixture of cell debris, macrophages, type I collagen, and fibroblasts, and is light in color compared to the luteum. The process is also known as luteal involution or luteolysis (*Rolaki et al., 2005*; *Stocco et al., 2007*; *Tilly, 2001*). Such a process stops the production of progesterone and triggers another cycle of ovulation. When mammals age, their follicles and corpora lutea are gradually lost, and the ovary eventually becomes a corpus albicans-filled organ that ceases to produce eggs and hormones (*Perheentupa and Huhtaniemi, 2009*; *Tilly, 2001*).

In the current study, we report that RIPK3 specifically induces apoptosis instead of necroptosis in certain contexts and cell types. The apoptosis-inducing function of RIPK3 requires phosphorylation of the evolutionarily conserved serine 165/threonine 166 sites (mouse origin) on its kinase activation loop, resulting in the inactivation of its kinase activity and acquiring the ability to recruit RIPK1-FADD-caspase-8, which leads to activation of caspase-3 and apoptosis. The apoptosis vs. necroptosis function of RIPK3 seemed to be determined by the cellular level of a RIPK3 chaperone Hsp90/CDC37. We also found that the serine 165/threonine 166 phospho-RIPK3 signal specifically appeared in corpora lutea/albicans when mice reached more than 4 months of age and $PGF_{2\alpha}$ can specifically trigger this apoptotic pathway during the process of corpus luteum involution in a mouse hyper-ovulation model.

## Results

### Ectopic expression of RIPK3 in cultured MCF7 or human granulosa lutein cells activated apoptosis through the RIPK1-FADD-caspase-8 pathway

We engineered three Dox-inducible RIPK3 expressing cell lines: HeLa/TO-RIPK3, MCF7/TO-RIPK3, and KGN/TO-RIPK3, all of which do not express endogenous RIPK3. KGN is a human granulosa tumor cell line (*Nishi et al., 2001*). Surprisingly, we found that simple expression of RIPK3 in MCF7 or KGN cells caused cell death (*Figure 1A, Figure 1—figure supplement 1*), whereas the ectopic expression of RIPK3 in HeLa cells did not affect cell viability, similar to previous studies (*Sun et al., 2012*). Moreover, these RIPK3-expressing MCF7 and KGN cells did not undergo necroptosis when we treated the cells with TSZ (*TNFα* [T], a *S*mac mimetic [S], and a pan-caspase inhibitor *Z*-VAD-fml [Z]), whereas such treatment caused robust necroptosis in RIPK3-expressing HeLa cells (*Figure 1B*).

To determine what kind of cell death was triggered by RIPK3 in MCF7 and KGN cells, we used caspase inhibitor z-VAD-fmk, RIPK1 inhibitor RIPA-56 (*Li et al., 2017*), and MLKL inhibitor NSA (*Sun et al., 2012*) to co-treat these cells with Dox. Cell death induced by Dox could be blocked by caspase inhibitor z-VAD-fmk (*Figure 1C, Figure 1—figure supplement 1*) but not by MLKL inhibitor NSA, nor RIPK1 kinase inhibitor RIPA-56 (*Figure 1C, Figure 1—figure supplement 1*), indicating that the observed RIPK3-mediated cell death was apoptotic. However, this form of apoptosis did not signal through the classic intrinsic mitochondrial pathway as the loss of cell viability through this pathway could not be blocked by caspase inhibitors nor was the apoptosis triggered by the TNF receptor family, which would be RIPK1 kinase-dependent.

To further explore the molecular mechanism of this apoptosis pathway, we expressed a previously reported kinase-dead mutant (K50A) of RIPK3 in MCF7 cells and found that this kinase-dead mutant RIPK3(K50A) did not cause cell death (*Figure 1D*). Interestingly, the expression of this kinase-dead mutant RIPK3(K50A) plus RIPK3 kinase inhibitor GSK'872 resulted in robust apoptosis, similar to what was described by Mandal et al. (*Figure 1D*; *Mandal et al., 2014*). Moreover, the mutations at the RHIM domain of RIPK3 known to disrupt RHIM-RHIM interaction also caused complete loss of cell death induction, indicating that RHIM domain interaction, possibly with another RHIM domain-containing protein, is required for apoptotic induction by RIPK3 (*Figure 1D*).

RIPK3's dependence on the RHIM domain but not its kinase activity suggested that the observed apoptosis might be through the RIPK1-FADD-caspase-8 pathway, as was seen in apoptosis induced by the D161N mutant RIPK3, a kinase-dead mutant of RIPK3 that causes spontaneous apoptosis and prenatal lethality in mice carrying this knock-in mutation (*Newton et al., 2014*). We therefore immunoprecipitated the Flag-tagged RIPK3 and probed the immune-precipitate with antibodies against RIPK1, caspase-8, and FADD. Indeed, all three proteins were co-immunoprecipitated with RIPK3. The amount of RIPK1 and caspase-8 associated with RIPK3 increased when a caspase inhibitor was present, indicating the complex had increased stability when caspase-8 activity was blocked, presumably by preventing caspase-8-mediated cleavage of RIPK1, which serves as an adaptor protein between RIPK3 (through the RHIM domain) and FADD (through the DEAD domain) (*Figure 2A*; *Mandal et al., 2014*; *Newton et al., 2014*; *Sun et al., 1999*). When we used the CRISPR/Cas9 to knock out either *RIPK1*, *caspase-8*, or *FADD* gene in MCF7/TO-RIPK3 cells, we found that cell death upon addition of Dox was blocked entirely (*Figure 2B–D*). Interestingly, different from the previous report (*Mandal et al., 2014*), knocking out the *cFLIP* gene did not block RIPK3-induced apoptosis in MCF7 cells but rather enhanced it (*Figure 2E*).

### RIPK3 kinase-dependently phosphorylates serine 164/threonine 165 in apoptotic cells

The differential apoptotic response of HeLa cells and MCF7/KGN cells to RIPK3 expression prompted us to search for the differences in RIPK3 protein in these cells. To this end, we immunoprecipitated Flag-RIPK3 from HeLa cells and MCF-7 cells and performed a detailed mass spectrometry analysis. We found two amino acid residues at the activation loop of the RIPK3 kinase domain, serine 164 and threonine 165, that were specifically phosphorylated in MCF7 cells but not in HeLa cells (*Figure 3A*). To validate this phosphorylation event, we generated a monoclonal antibody against the phospho-S164/T165 peptide. This phospho-specific antibody only recognized RIPK3

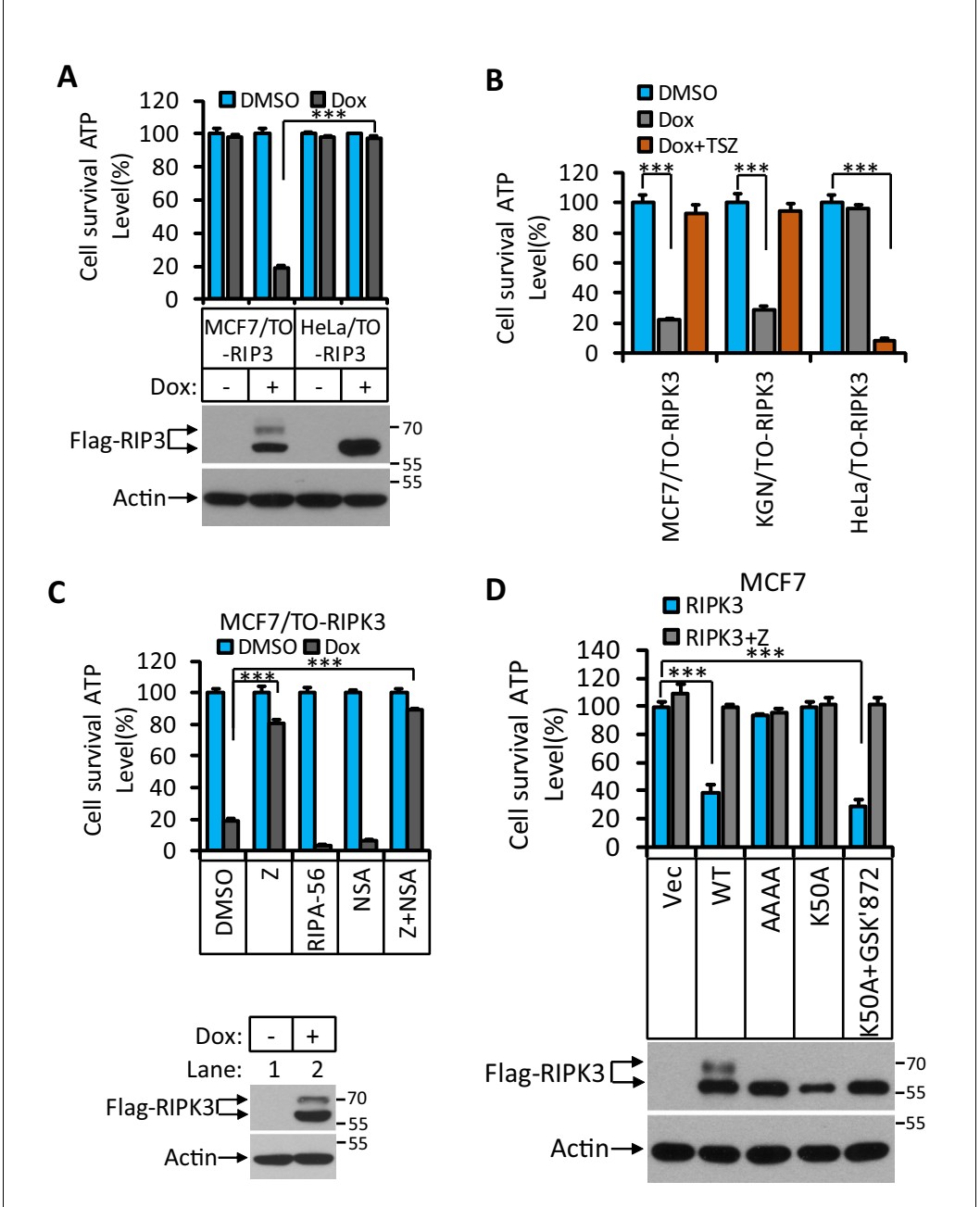

**Figure 1.** RIPK3-induced apoptosis in MCF7 and KGN cells. (**A**) Cultured MCF7/TO-RIPK3 and HeLa/TO-RIPK3 cells were treated with DMSO or Dox (1 µg/ml) induction for 36 hr. Cell viability was determined by measuring cellular ATP levels (upper panel). The data are represented as the mean ± SD of triplicate wells. \*\*\*p<0.001. p-values were determined by two-sided unpaired Student's *t*-tests. The cell lysates were analyzed by western blotting using antibodies against RIPK3 or β-actin (lower panel). (**B**) Cultured MCF7/TO-RIPK3, KGN/TO-RIPK3, and HeLa/TO-RIPK3 cells were treated with DMSO, Dox, or Dox plus TSZ for 36 hr. Cell viability was determined by measuring cellular ATP levels. The data are represented as the mean ± SD of triplicate wells. \*\*\*p<0.001. p-values were determined by two-sided unpaired Student's *t*-tests. (**C**) Cultured MCF7/TO-RIPK3 cells were treated with DMSO or Dox, plus the indicated agents for 36 hr. Cell viability was determined by measuring cellular ATP levels (upper panel). The data are represented as the mean ± SD of triplicate wells. \*\*\*p<0.001. p-values were determined by two-sided unpaired Student's *t*-tests. The cell lysates were analyzed by western blotting using antibodies against RIPK3 or β-actin (lower panel). 20 µM Z, pan-caspase inhibitor z-VAD; 2 µM RIPA-56, RIPK1 inhibitor; 2 µM NSA, MLKL inhibitor. (**D**) Cultured MCF7 cells were infected with lentiviruses encoding RIPK3(WT), RIPK3(AAAA), RIPK3(K50A), and RIPK3(K50A)+GSK'872 plus Z for 36 hr. Cell viability was determined by measuring

*Figure 1 continued on next page*

**Figure 1 continued**

cellular ATP levels (upper panel). The data are represented as the mean ± SD of triplicate wells. ***p<0.001.
p-values were determined by two-sided unpaired Student's *t*-tests. The lysates were measured by western blotting
using antibodies against RIPK3 or β-actin as indicated (lower panel). GSK'872, RIPK3 inhibitor.
The online version of this article includes the following figure supplement(s) for figure 1:

**Figure supplement 1.** RIPK3-induced apoptosis in human granulosa lutein cells (KGN).

protein expressed in KGN or MCF7 cells but not in HeLa cells (*Figure 3B*). We further validated our
monoclonal antibody's specificity by expressing a RIPK3 protein bearing phosphorylation-resistant
mutations S164A/T165A in MCF and KGN cells. The expressed S164A/T165A RIPK3 was then ana-
lyzed by western blotting with antibodies against RIPK3 or phospho-S164/T165 RIPK3. As shown in
*Figure 3—figure supplement 1A, B*, Dox-induced wild-type RIPK3 was recognized by anti-RIPK3
and anti-phospho-S164/T165 RIPK3 antibodies in both MCF7/TO-RIPK3 and KGN/TO-RIPK3 cells.
The phosphorylation-resistant S164A/T165A mutant protein could only be recognized by the anti-
RIPK3 antibody but not the anti-phospho-S614/T165 antibody. Notably, this anti-phospho-S164/
T165 antibody did not recognize any non-specific protein band across the entire set of protein size

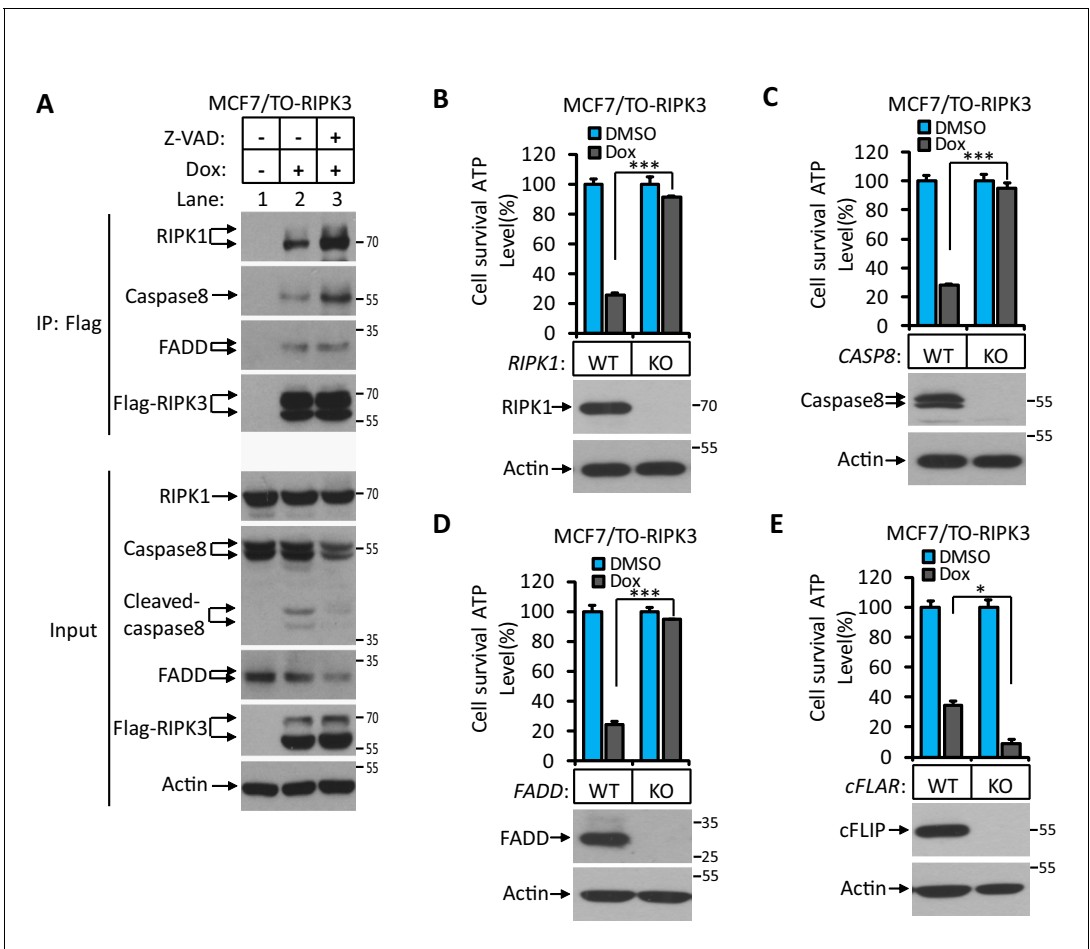

**Figure 2.** RIPK3-induced apoptosis was dependent on RIPK1, FADD, and caspase-8. (**A**) Cultured MCF7/TO-RIPK3 cells were treated with DMSO or
Dox plus the indicated agent for 24 hr. The cells were then harvested, and RIPK3 was immunoprecipitated from the cell lysates using anti-Flag resin.
The cell lysates and immunocomplexes were analyzed by western blotting using antibodies as indicated. (**B–E**) Cultured MCF7/TO-RIPK3 (wild type
[WT], *RIPK1*$^{-/-}$, *Caspase8*$^{-/-}$, *FADD*$^{-/-}$, and *cFLIP*$^{-/-}$) cells were treated with DMSO or Dox induction for 36 hr. Cell viability was determined by measuring
cellular ATP (upper panel). The data are represented as the mean ± SD of triplicate wells. *p<0.05, ***p<0.001. p-values were determined by two-
sided unpaired Student's *t*-tests. The cell lysates were analyzed by western blotting using antibodies against RIPK1, caspase-8, FADD, cFLIP, or β-actin
(lower panel). Five independent knockout clones were test in each gene.

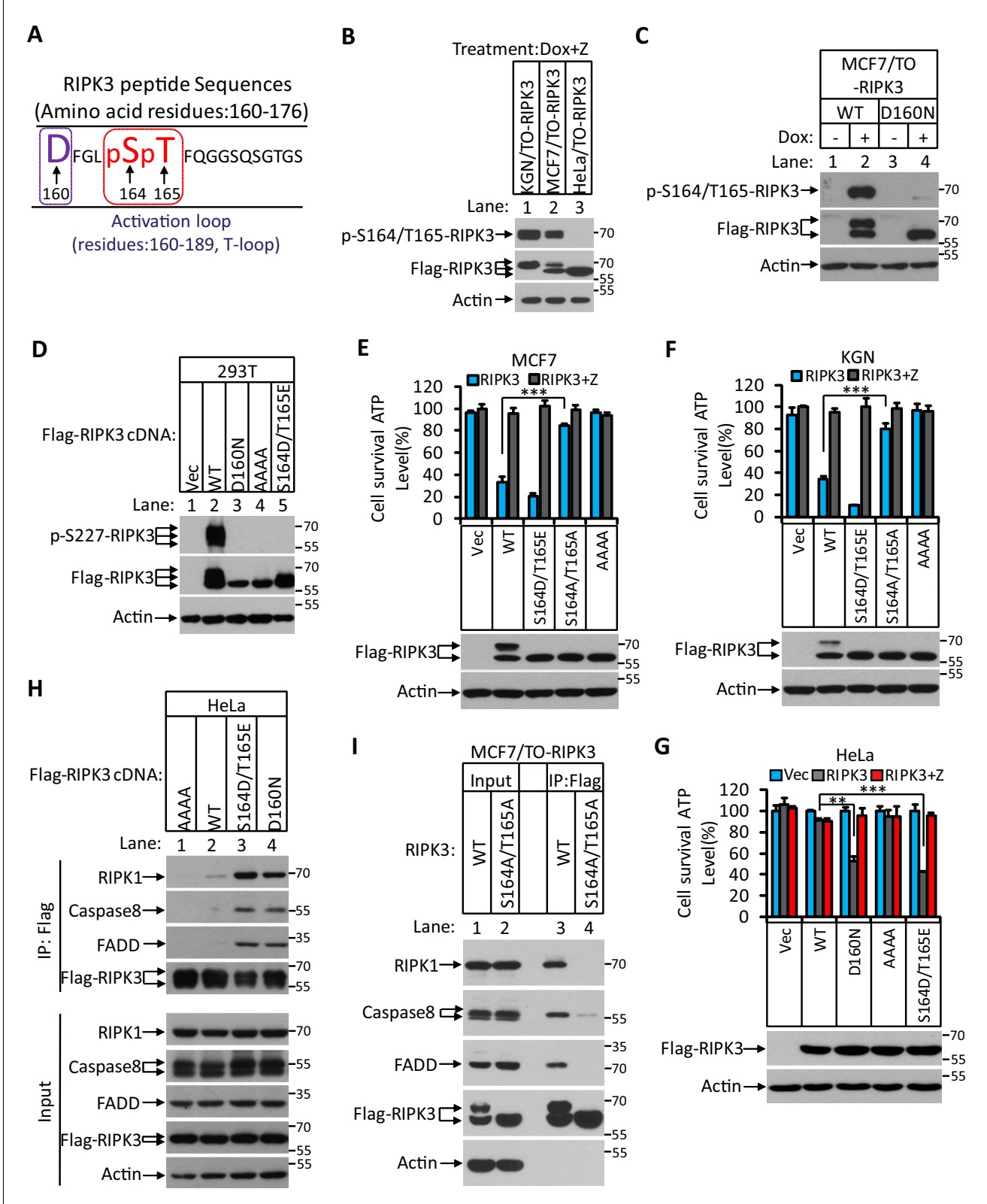

**Figure 3.** RIPK3-mediated apoptosis is dependent on the auto-phosphorylation of the S164/T165 sites. (**A**) Cultured MCF7/TO-RIPK3 and HeLa/TO-RIPK3 cells were treated with Dox plus z-VAD for 24 hr. RIPK3 was immunoprecipitated from the cell lysates using anti-Flag resin. The RIPK3 bands were excised and subjected to mass spectrometry analysis. RIPK3-specific phosphorylation site in MCF7/TO-RIPK3 cells is highlighted in red. (**B**) Cultured KGN/TO-RIPK3, MCF7/TO-RIPK3, and HeLa/TO-RIPK3 cells were treated with Dox plus z-VAD for 24 hr. The lysates were analyzed by western blotting

*Figure 3 continued on next page*

Figure 3 continued

using antibodies against the phospho-serine 164/threonine 165 of RIPK3, Flag (RIPK3), and β-actin as indicated. (C) Cultured MCF7 stably transfected with either wild-type RIPK3 (WT) or kinase-dead mutant (D160N) cells under the control of Dox-inducible promoter were treated with DMSO(-) or Dox plus z-VAD for 24 hr. The lysates were analyzed by western blotting using antibodies against the phospho-serine 164/threonine 165 RIPK3, Flag (RIPK3), and β-actin as indicated. (D) Cultured 293T cells were transfected with Vector (Vec), RIPK3(WT), RIPK3(D160N), RIPK3(AAAA) (RIPK3-AAAA, residues 459–462 mutated to AAAA), and RIPK3(S164D/T165E) for 24 hr. The level of phospho-S227-RIPK3 and RIPK3 was measured by western blotting. (E, F) Cultured MCF7 (E) and KGN (F) cells were infected with lentiviruses encoding RIPK3(WT), RIPK3(S164D/T165E), RIPK3(S164A/T165A), and RIPK3(AAAA) plus z-VAD for 36 hr. Cell viability was determined by measuring cellular ATP levels (upper panel). The data are represented as the mean ± SD of triplicate wells. ***p<0.001. p-values were determined by two-sided unpaired Student's t-tests. The lysates were measured by western blotting using antibodies against RIPK3 or β-actin as indicated (lower panel). (G) Cultured HeLa cells were infected with lentiviruses encoding RIPK3(WT), RIPK3 (D160N), RIPK3(AAAA), and RIPK3(S164D/T165E) plus z-VAD for 36 hr. Cell viability was determined by measuring cellular ATP levels (upper panel). The data are represented as the mean ± SD of triplicate wells. **p<0.01, ***p<0.001. p-values were determined by two-sided unpaired Student's t-tests. The expressed RIPK3 in the cell lysates were measured by western blotting using antibodies against RIPK3 or β-actin as indicated (lower panel). Vector (Vec, control viruses) (H) Cultured HeLa cells were transfected with Flag-tagged RIPK3(WT), RIPK3(D160N), and RIPK3(S164D/T165E) for 24 hr. RIPK3 was immunoprecipitated using anti-Flag resin. The lysates and immunocomplexes were analyzed by western blotting using antibodies against RIPK1, caspase-8, FADD, and RIPK3 as indicated. (I) Cultured MCF7/TO-RIPK3 and MCF7/TO-RIPK3(S164A/T165A) cells were treated with Dox plus Z for 24 hr. The cells were then harvested, and RIPK3 was immunoprecipitated from the cell lysates using anti-Flag resin. The cell lysates and immunocomplexes were analyzed by western blotting using antibodies as indicated.

The online version of this article includes the following figure supplement(s) for figure 3:

**Figure supplement 1.** Characterization of RIPK3 auto-phosphorylation sites.

**Figure supplement 2.** The phosphorylation site of RIPK3 is conserved among different mammalian species.

markers (15–250 kDa), nor in cells without Dox addition to their media. Additionally, the expression of two kinase-dead mutants RIPK3 D160N and RIPK3 K50A (equivalent of mouse D161N and K51A, respectively) failed to generate the phospho-S164/T165 signal (*Figure 3C, Figure 3—figure supplement 1C*). Therefore, the RIPK3-induced apoptosis in KGN and MCF7 cells requires serine 164/threonine 165 phosphorylation of RIPK3, an event dependent on its intact kinase domain.

## Phosphorylation of serine 164 inactivates the kinase activity of RIPK3 and activates apoptosis

To explore the functional significance of this phosphorylation event, we made phospho-mimic point mutations in these two sites and assayed the kinase activity of mutant RIPK3 by probing the serine 227 phosphorylation, which is known to be carried out by RIPK3 kinase activity and is critical for recruiting its necroptosis executing substrate MLKL (*Li et al., 2015*; *Sun et al., 2012*). After transiently transfecting the cDNA of mutant RIPK3 in human embryonic kidney 293T cells, we observed that RIPK3 bearing phospho-mimic S164D/T165E mutations failed to auto-phosphorylate the serine 227 site, similar to the kinase-dead mutant D160N, whereas wild-type RIPK3 readily phosphorylated serine 227 (*Figure 3D*).

We subsequently measured cell death after introducing these phospho-mimic or phosphorylation-resistant mutations in MCF7 and KGN cells. As shown in *Figure 3E, F*, introducing phospho-mimic mutations S164D/T165E increased apoptosis in MCF7 and KGN cells compared to wild-type RIPK3, whereas the phosphorylation-resistant mutant S164A/T165A lost the ability to induce apoptosis, similar to the RHIM domain mutant. Remarkably, in HeLa cells, the phospho-mimic mutant S164D/T165E gained the ability to induce apoptosis similar to the kinase-dead mutant D160N (*Figure 3G*), indicating that phospho-mimic mutations activated RIPK3's apoptosis-inducing function that is normally silenced in HeLa cells. Since the phospho-mimic mutations inactivate RIPK3 kinase activity (*Figure 3D*), this S164D/T165E mutant was no longer able to induce necroptosis in HeLa cells when treated with necroptosis stimuli TSZ (*Figure 3—figure supplement 1D*). The phosphorylation-resistant mutant S164A/T165A still allowed HeLa cells to respond to necroptosis induction, though the apoptosis-inducing function was lost when the mutant RIPK3 was expressed in KGN and MCF7 cells (*Figure 3—figure supplement 1D*). Thus, the differential responses of RIPK3 in HeLa cells compared to KGN and MGF7 cells are due to its inability to carry out auto-phosphorylation at the serine 164/threonine 165 sites.

We further narrowed down the effect of serine 164 and threonine 165 phosphorylation by generating individual phospho-mimic or phosphorylation-resistant mutants at serine S164 or threonine 165

sites and measured their kinase activity and apoptosis-inducing ability. The phospho-mimic S164E, but not T165E, mutant lost necroptosis responsiveness and gained the ability to induce apoptosis in HeLa cells, while phosphorylation-resistant S164A, but not T165A, lost its apoptosis-inducing activity in KGN and MCF7 cells (*Figure 3—figure supplement 1D–G*). Together, these results indicated that RIPK3 auto-phosphorylation on the serine S164 site is a critical event that inactivates RIPK3 kinase activity while gaining apoptosis-inducing activity in KGN and MCF7 cells.

## Serine 164 phosphorylation allows RIPK3 binding to RIPK1-FADD-caspase-8

To study the mechanism by which phospho-S164/T165 RIPK3 activates the RIPK1/FADD/caspase-8 pathway, we ectopically expressed Flag-RIPK3 (wild type), phospho-mimic RIPK3 (S164D/T165E), and RIPK3 (D160N) kinase-dead mutants, as well as RIPK3(AAAA) RHIM domain mutants in HeLa cells and performed immunoprecipitation with an anti-Flag antibody. The immune-complexes were probed with antibodies against RIPK1, FADD, and caspase-8. As shown in *Figure 3H*, very little RIPK1, FADD, or caspase-8 were co-precipitated with wild-type or RHIM mutant RIPK3, consistent with its inability to induce apoptosis (*Figure 3H*, lanes 1 and 2). In contrast, two kinase-dead mutants, either the artificial D160N or phospho-mimic S164D/T165E, showed robust binding with RIPK1, FADD, and caspase-8 (*Figure 3H*, lanes 3 and 4). Consistent with the cell death result, the phosphorylation-resistant mutant S164A/T165A lost the ability to bind with RIPK1, FADD, and caspase-8 (*Figure 3I*).

Consistent with the finding that the serine 164 phosphorylation is most critical for RIPK3 apoptosis-inducing activity, the phospho-mimic S164E mutant expressed in HeLa cells was found to associate with RIPK1, FADD, and caspase-8, as did the S164D/T165E double mutation (*Figure 3—figure supplement 1H*, lane 4), whereas T165E alone could not bind to these proteins (*Figure 3—figure supplement 1H*, lane 2). Collectively, these findings indicate that RIPK3 auto-phosphorylation on serine S164 allows RIPK3 to recruit RIPK1 to form the RIPK3/RIPK1/FADD/caspase-8 complex, in which caspase-8 is activated to drive apoptosis.

## Hsp90/CDC37 chaperone determines the necroptotic or apoptotic function of RIPK3 kinase

We have previously observed that in order for RIPK3 to signal to necroptosis the protein needs to be correctly folded with the aid of a molecular chaperone consisting of Hsp90 and CDC37 (*Li et al., 2015*). In HT29 and HeLa cells in which RIPK3 signals primarily to necroptosis, we observed that the level of Hsp90/CDC37 was higher compared to MCF7 and KGN cells, where RIPK3 predominantly causes apoptosis (*Figure 4A*). Indeed, a human Hsp90 inhibitor 17AAG potently blocked necroptosis induced by TSZ in HeLa-RIPK3 cells, whereas 17AAG did not block RIPK3-induced apoptosis in MCF7/TO-RIPK3 or KGN/TO-RIPK3 cells (*Figure 4B, C*). When we knocked down Hsp90 using shRNA or inhibits Hsp90 using 17AAG, after Dox induce RIPK3 expression, we saw the switch of the cellular necroptosis response to apoptosis (*Figure 4D, Figure 4—figure supplement 1A*). Consistently, the serine 164/threonine 165 phosphorylation of RIPK3 also occurred after the inhibition of Hsp90 (*Figure 4E*). Reciprocally, when Hsp90 and CDC37 were ectopically expressed in MCF7/TO-RIPK3 cells to a level matched or exceeding that in HeLa cells, the serine 164/threonine 165 phosphorylation of RIPK3 no longer occurred, and RIPK3-mediated apoptosis induction mostly ceased (*Figure 4F, G*). Consistent with the cell death result, when the inhibitor of Hsp90 17AAG was present during RIPK3 induction in HeLa/TO-RIPK3 cells, the RIPK3 formed a complex with RIPK1, FADD, and caspase-8 (*Figure 4H*).

Since serine 164/threonine 165 appeared to be at a critical regulatory region in RIPK3 on which phosphorylation switches its function from necroptosis to apoptosis, we checked this region among different mammalian species. As shown in *Figure 3—figure supplement 2A*, this region is conserved with no changes from amino acid residues 160–174 among RIPK3s from human, chimpanzee, rat, mouse, and bovine origin (*Xie et al., 2013*). The corresponding phospho-mimic mutations of mouse RIPK3(S165D/T166E), like RIPK3(D161N) kinase-dead mutants (*Newton et al., 2014*), failed to auto-phosphorylate the serine 232 site, the equivalent of the human serine 227 site (*Figure 3—figure supplement 2B*). Furthermore, these mutant RIPK3 proteins induced apoptosis in mouse L929 (*Ripk3<sup>−/−</sup>*) cells after ectopic expression but not necroptosis when treated with the necroptosis

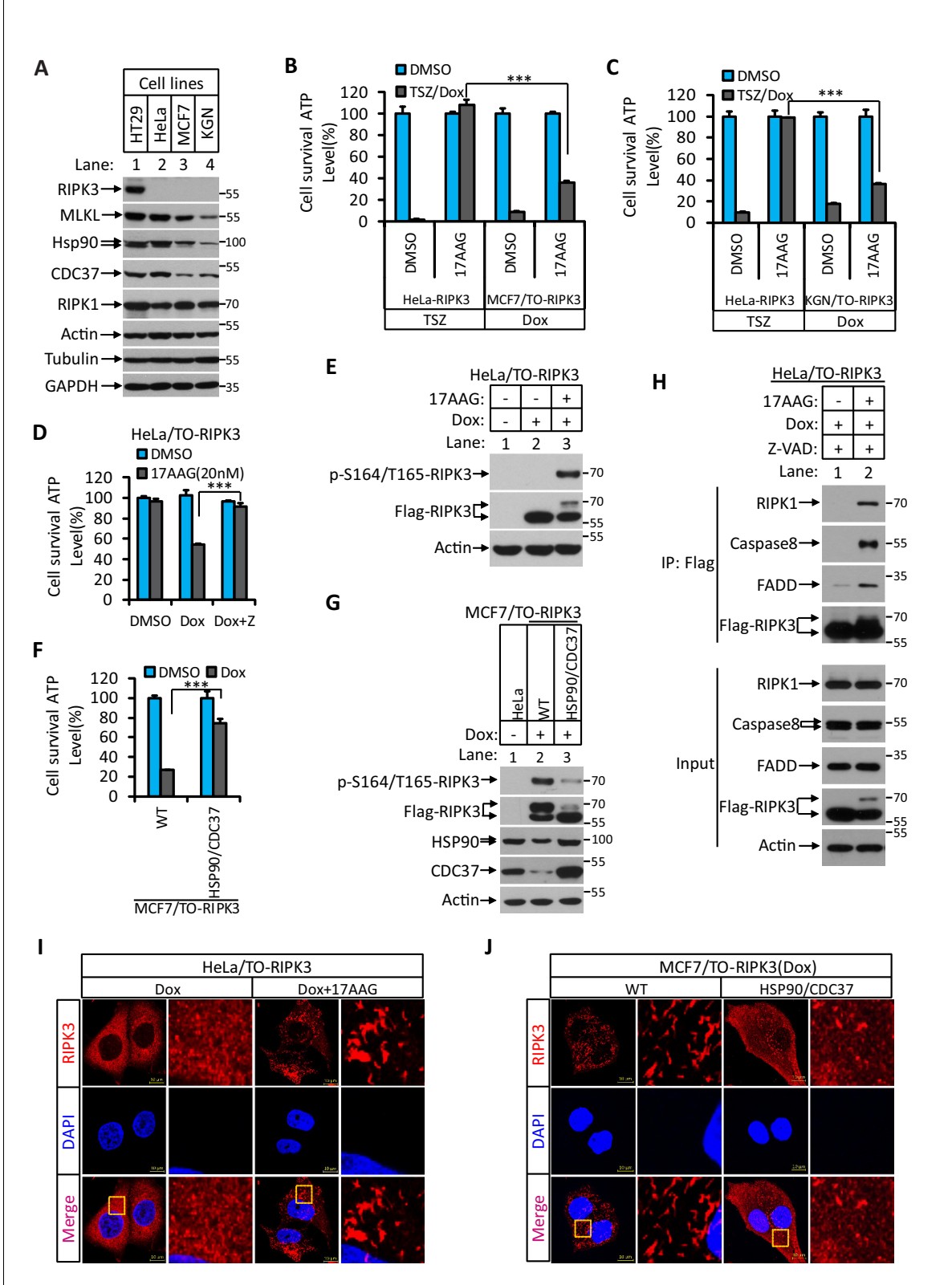

**Figure 4.** Hsp90/CDC37 chaperone determined the apoptotic and necroptotic function of RIPK3 kinase. (**A**) The cell lysates from cultured HT29, HeLa, MCF7, and KGN cells were analyzed by western blotting using antibodies as indicated. (**B, C**) Cultured HeLa-RIPK3, MCF7/TO-RIPK3, and KGN/TO-RIPK3 cells were treated with the indicated stimuli for 36 hr. Cell viability was determined by measuring cellular ATP levels. The data are represented as the mean ± SD of triplicate wells. ***p<0.001. p-values were determined by two-sided unpaired Student's *t*-tests. 17AAG, Hsp90 inhibitor. (**D, E**) HeLa/
*Figure 4 continued on next page*

*Figure 4 continued*

TO-RIPK3 cells were treated with the indicated stimuli for 36 hr. Cell viability was determined by measuring cellular ATP levels in (D). The data are represented as the mean ± SD of triplicate wells. ***p<0.001. p-values were determined by two-sided unpaired Student's *t*-tests. 24 hr after treatment, the cell lysates were analyzed by western blotting using antibodies against phospho-serine 164/threonine 165 of RIPK3, RIPK3, and β-actin as indicated in (E). (F, G) Cultured MCF7/TO-RIPK3 cells co-transfected with HSP90 and CDC37 as indicated were treated with DMSO or Dox for 36 hr. Cell viability was determined by measuring cellular ATP levels in (F). The data are represented as the mean ± SD of triplicate wells. ***p<0.001. p-values were determined by two-sided unpaired Student's *t*-tests. 24 hr after treatment, the cell lysates were analyzed by western blotting using antibodies against phospho-serine 164/threonine 165 of RIPK3, RIPK3, Hsp90, CDC37, and β-actin as indicated in (G). (H) Cultured HeLa/TO-RIPK3 cells were treated with Dox or Dox plus 17AAG for 24 hr. The cells were then harvested, and RIPK3 was immunoprecipitated from the cell lysates using anti-Flag resin. The cell lysates and immunocomplexes were analyzed by western blotting using antibodies as indicated. (I) Cultured HeLa/TO-RIPK3 cells were treated with Dox or Dox plus 17AAG for 24 hr. Immunofluorescence of the cells with Flag-RIPK3 (red) antibody. Counterstaining with DAPI (blue). Scale bar, 10 μm. Higher-power views (right panels) were acquired from the selected boxed areas from the left panel. (J) Cultured MCF7/TO-RIPK3 cells co-transfected with HSP90 and CDC37 as indicated were treated with Dox for 24 hr. Immunofluorescence of the cells with Flag-RIPK3 (red) antibody. Counterstaining with DAPI (blue). Scale bar, 10 μm. Higher-power views (right panels) were acquired from the selected boxed areas from the left panel. The online version of this article includes the following figure supplement(s) for figure 4:

**Figure supplement 1.** Hsp90/CDC37 chaperone determines the necroptotic or apoptotic function of RIPK3 kinase.
**Figure supplement 2.** RIPK3 form amyloid-like structure in MCF7 and KGN cells.

stimulus TSZ (*Figure 3—figure supplement 2C*). Previously, we noticed that when we used a clinical relevant dose of 17AAG (200 nM) to treat mouse cells, the necroptosis was augmented instead of inhibited like in human or rat cells (*Li et al., 2015*). We therefore suggested that the Hsp90/CDC37 chaperone might not work in mouse cells. Given the importance of this chaperone system in determining the apoptotic or necroptotic fate and the conservation of the apoptosis-causing phosphorylation site in mice, we explored the possibility that the previously tested concentration of human Hsp90 inhibitors was simply not enough to inhibit the mouse Hsp90. Indeed, when the higher dose of 17AAG was used, the compound dose-dependently blocked TSZ-induced necroptosis in L929 cells (*Figure 4—figure supplement 1B*). Moreover, when we knocked down Hsp90 using shRNA or inhibits Hsp90 using higher concentrations of 17AAG in L929 (*Ripk3<sup>-/-</sup>*)/TO-RIPK3 cells, we saw the switch of the cellular necroptosis response to apoptosis after Dox induced RIPK3 expression (*Figure 4—figure supplement 1C, D*). The serine 165/threonine 166 phosphorylation of RIPK3 also occurred after the inhibition of Hsp90 (*Figure 4—figure supplement 1E*). These results indicated that the observed Hsp90/CDC37-determined, RIPK3 serine 164/threonine 165 phosphorylation-driven apoptosis is conserved between human and mouse cells.

## RIPK3 forms a unique structure in MCF7 and KGN cells

RIPK3 is known to form diffused puncta within the cytosol (*He et al., 2009*). Indeed, when the HeLa/TO-RIPK3 cells were treated with Dox plus TSZ, we observed expressed RIPK3 in diffused puncta in the cytosol (*Figure 4—figure supplement 2A*). Interestingly, when we performed a similar analysis of RIPK3 distribution in MCF7/TO-RIPK3 and KGN/TO-RIPK3 cells, the RIPK3 formed a high order of structures distinct from the diffused punctae in HeLa or HT29 cells (*Figure 4—figure supplement 2A, B*). Furthermore, the phospho-S164/T165-RIPK3 signal appeared co-localized with RIPK3 in MCF7 cells, but no such signal was seen in HeLa cells (*Figure 4—figure supplement 2C*). The formation of these high-order structures and the appearance of the phospho-S164/T165-RIPK3 signal were not affected when *RIPK1*, *caspase-8*, or *FADD* genes were knocked out (*Figure 4—figure supplement 2D, E*), indicating that it was RIPK3, not its downstream apoptotic effectors, that was responsible for forming these high-order intracellular structures.

To test whether this high-order structure of RIPK3 is associated with its ability to induce apoptosis, we treated the HeLa/TO-RIPK3 cells with Dox plus the Hsp90 inhibitor 17AAG and found that the diffused punctae of RIPK3 were now switched to the high-order structure (*Figure 4I*). Reciprocally, when Hsp90 and CDC37 were ectopically overexpressed in MCF7/TO-RIPK3 cells, the high-order structure of RIPK3 became diffused punctae throughout the cytosol of the cells (*Figure 4J*). These findings collectively indicated that RIPK3 forms distinctive cellular structures when signaling to apoptosis or necroptosis, depending on the chaperone Hsp90/CDC37.

## *Ripk3*<sup>*S165D-T166E/S165D-T166E*</sup> mice die within a month after birth due to spontaneous apoptosis in multiple tissues

We subsequently sought genetic proof that the phosphorylation of RIPK3 at its serine 165/threonine 166 sites (equivalent to human serine 164/threonine 165 sites) drives apoptosis in vivo. To this end, we generated *Ripk3* knock-in mice with Ser$^{165}$Thr$^{166}$ mutated to Ala-Ala or Asp-Glu (*Figure 5—figure supplement 1A, B, E*) to either block or mimic the phosphorylation on these sites (*Figure 5—figure supplement 1A, B, F*). Interestingly, the *Ripk3*$^{S165A-T166A/S165A-T166A}$ homozygous mice were viable and appeared normal (*Figure 5—figure supplement 1C, D, F*). *Ripk3*$^{S165D-T166E/S165D-T166E}$ homozygous mice, on the other hand, are smaller in size and all died within 1 month after birth (*Figure 5A–C*). Anatomical analysis of these *Ripk3*$^{S165D-T166E/S165D-T166E}$ mice showed abnormal large intestine, small intestine, lung, spleen, and kidney, while other major organs, including brain, cerebellum, heart, and liver, appeared normal (*Figure 5D, Figure 5—figure supplement 1G*). Staining for cleaved caspase-3 revealed significantly increased apoptosis in the large intestine, small intestine, lung, and spleen (*Figure 5E, F*). The *Ripk3*$^{S165D-T166E/+}$ heterozygous mice, which were expected to make half the amount of RIPK3 S165D/T166E protein, are viable. One explanation is that there must be a threshold above which the RIPK3 S165D/T166E protein becomes lethal. Alternatively, the wild-type RIPK3 protein might prevent RIPK3(S165D/T166E) activation by forming a dimer with the mutant protein.

Consistent with what was seen in HeLa cells, in which the phospho-mimic mutations S164D/T165E of RIPK3 were unable to signal necroptosis, the bone marrow-derived macrophages (BMDM) from *Ripk3*$^{S165D-T166E/S165D-T166E}$ mice were resistant to necroptosis induction by TSZ (*Figure 5G*). Conversely, the BMDM from *Ripk3*$^{S165A-T166A/S165A-T166A}$ mice readily underwent necroptosis when treated with TSZ, as did the BMDM from wild-type mice (*Figure 5—figure supplement 1H*). It is worth noting that when we cultured BMDM from *Ripk3*$^{S165D-T166E/S165D-T166E}$ mice we needed to include the caspase inhibitor z-VAD-fmk in the culture medium to prevent spontaneous cell death (see Materials and methods).

## PGF$_{2\alpha}$ elevates the expression of RIPK3 in luteal granulosa cells to promote luteum regression

Among all RIPK3-expressing tissues, RIPK3 expression levels in the ovary were the most notable, following a reversed bell curve pattern through the lifetime of the mice (*Figure 6A*). In early life and young adulthood, RIPK3 expression was restricted to the follicle granular cells around the egg (*Figure 6B*). In later life, RIPK3 protein starts to appear in the corpus luteum and albican (*Figure 6B*). The second wave of expression coincides with the elevation of prostaglandin F2alpha (PGF$_{2\alpha}$), which has been reported to have a role in apoptosis induction during luteal involution (*Parkening et al., 1985*). Indeed, when we measured the PGF$_{2\alpha}$ levels in the mouse ovary, we saw a significant age-dependent increase as mice aged from 4 months to 10 and 16 months, coinciding with the age-dependent elevation of RIPK3 (*Figure 6C*). Moreover, when we co-stained 12-month-old RIPK3-Flag knock-in ovaries with anti-Flag and prostaglandin F receptor (PTGFR), we noticed that the PTGFR signal was primarily located in corpus lutea and corpus albicans but not in the follicles and co-localized with that of RIPK3 (*Figure 6D, Figure 6—figure supplement 1A, B*).

We thus used the same anti-phospho-S164/T165 antibody to probe ovary extracts from mice of different ages by western blotting. We found no phosphorylation signal in young (2-month-old) ovaries, despite the RIPK3 protein band being readily detectable (*Figure 6E*). The signal began to appear at 4 months and gradually increased when mice advanced in age (*Figure 6E*). To locate this phosphorylation signal in aged ovaries, we used this antibody for immunohistochemical analysis on 4-, 8-, and 12-month-old ovaries. Consistent with the western blotting, there was minimal signal in the young (4-month-old) ovary. There were patches of signal in 8-month-old corpus luteum and albican, and the signal became prominent and clustered in the corpus albicans in 12-month-old ovary (*Figure 6F*, middle panels). In contrast, no such phospho-RIPK3 signal was seen in 12-month-old ovaries of RIPK3 knockout mice, further confirming the specificity of this monoclonal antibody (*Figure 6F*).

As RIPK3 induce apoptosis or necroptosis depending on the chaperone Hsp90/CDC37 level, we thus measured the level of Hsp90/CDC37 in granulosa cells of follicles and corpus lutea/albicans of 8-month-old mouse ovaries using antibodies against these two proteins. Both antibodies stained

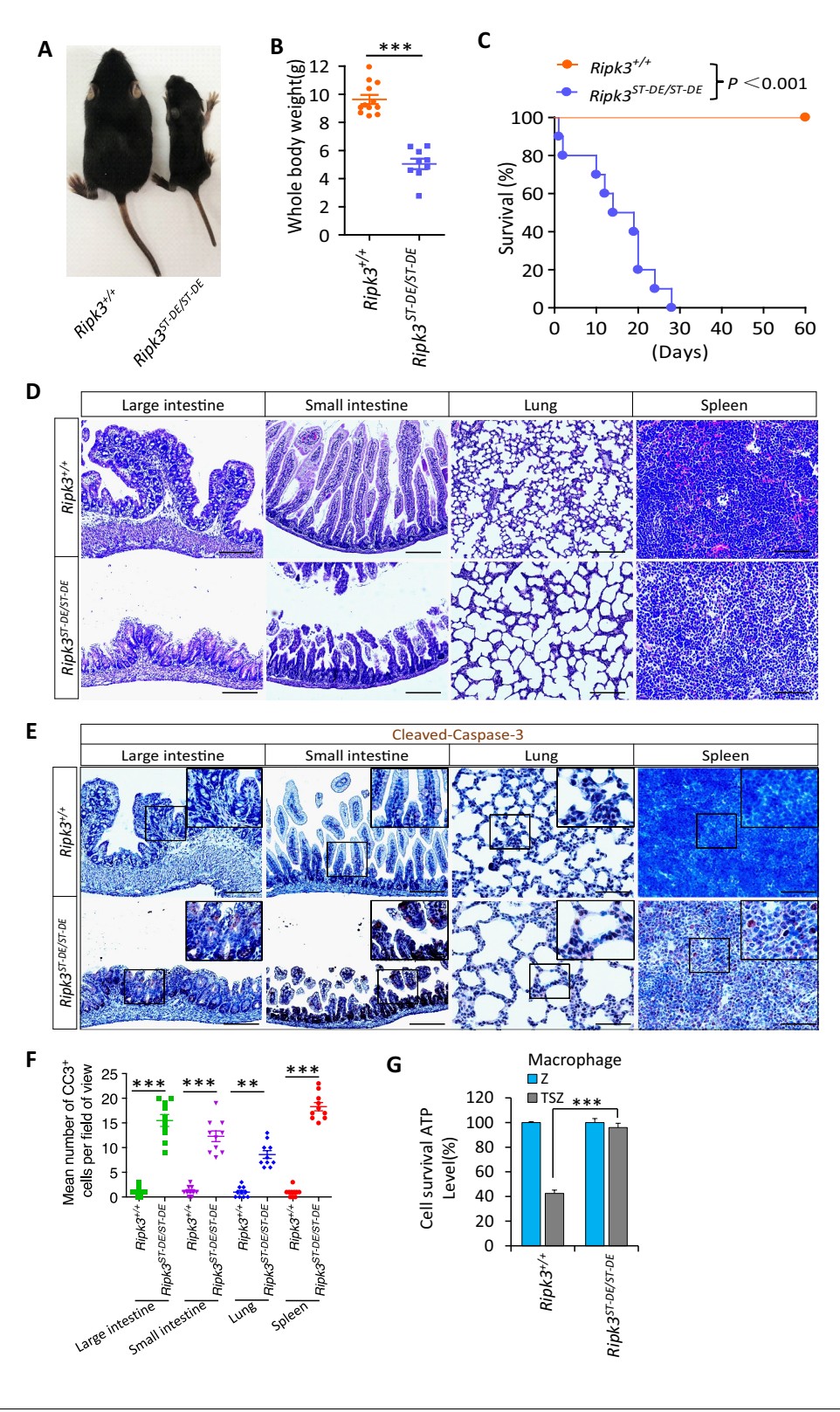

**Figure 5.** *Ripk3^{S165D-T166D/S165D-T166D}* (*Ripk3^{ST-DE/ST-DE}*) mice die within 1 month after birth. (**A, B**) Macroscopic features (**A**) and body weights (**B**) of *Ripk3^{+/+}* and *Ripk3^{ST-DE/ST-DE}* littermate mice at 14 days of age (n ≥ 9). The result from each individual animal is presented as an indicated dot. ***p<0.001. p-values were determined by two-sided unpaired Student's *t*-tests. (**C**) Kaplan–Meier plot of survival of *Ripk3^{+/+}* and *Ripk3^{ST-DE/ST-DE}* littermate mice (n = 10 for each genotype) after birth within 2 months. ***p<0.001. p-values were determined by two-sided unpaired Student's *t*-tests.
*Figure 5 continued on next page*

*Figure 5 continued*

(D) Histological analysis of large intestine, small intestine, lung, and spleen of $Ripk3^{+/+}$ and $Ripk3^{ST-DE/ST-DE}$ littermate mice (n = 5) at 14 days of age. Scale bar, 20 µm. (E, F) Representative immunohistochemistry (IHC) images of the large intestine, small intestine, lung, and spleen of $Ripk3^{+/+}$ and $Ripk3^{ST-DE/ST-DE}$ littermate mice (n = 5, 14 days) stained with a cleaved-caspase-3 (C–C3) antibody in (E). C-C3-positive cells were counted in two fields per organ and quantified in (F). Scale bar, 10 µm. Data represent the mean ± s.e.m. **p<0.01, ***p<0.001. p-values were determined by two-sided unpaired Student's $t$-tests. (G) Cell viability measurement of bone marrow-derived macrophages from the $Ripk3^{+/+}$ and $Ripk3^{ST-DE/ST-DE}$ littermate mice (n = 3, 14 days) after treatment with the indicated Z-VAD or necroptosis stimuli for 24 hr. Cell viability was determined by measuring cellular ATP levels. The data are represented as the mean ± SD of triplicate wells. ***p<0.001. p-values were determined by two-sided unpaired Student's $t$-tests. .

The online version of this article includes the following figure supplement(s) for figure 5:

**Figure supplement 1.** Generation of $Ripk3^{ST-DE/ST-DE}$ and $Ripk3^{ST-AA/ST-AA}$ mice.

**Figure supplement 2.** Hsp90/CDC37 chaperone protein levels were low in corpus luteum and corpus albicans.

follicle granulosa much more strongly than corpus lutea and albicans, whereas RIPK3 staining signal was at a similar level in both, indicating that the relative level of Hsp90/CDC37 vs. RIPK3 is higher in follicles than corpus lutea and albicans (*Figure 5—figure supplement 2*). Collectively, these results suggested that RIPK3-induced apoptosis only happens in aged corpus lutea and albicans might be due to relative lower levels of Hsp90/CDC37 chaperone in these regions of the ovary.

PGF$_{2\alpha}$ is known to stimulate the Raf/MEK1/mitogen-activated protein kinase signaling pathway to induce luteal regression (*Chen et al., 1998*; *Rolaki et al., 2005*; *Wang et al., 2003*; *Yadav et al., 2002*). In mice, knockout caspase-3 blocks PGF$_{2\alpha}$-induced regression of the corpus luteum (*Carambula et al., 2002*; *Carambula et al., 2003*). Those observations suggested that PGF$_{2\alpha}$ might induce RIPK3 expression by stimulating the Raf/MEK1/mitogen-activated protein kinase signaling pathway, thus causes RIPK3-mediated apoptosis. To test if PGF$_{2\alpha}$ could directly induce RIPK3 expression, we isolated primary granulosa lutein cells from young (3-month-old) mouse ovaries and treated them with a synthetic PGF$_{2\alpha}$ analog dinoprost tromethamine (DT) (*Chebel et al., 2007*). We then observed the time- and dose-dependent induction of RIPK3 protein, a pattern that overlapped with the appearance of phospho-ERK/MEK signal, an indication of PGF2 receptor activation (*Figure 7A, B*). The addition of two MEK kinase inhibitors PD-98059 or U0126 blocked RIPK3 induction, confirming that the RIPK3 induction by PGF$_{2\alpha}$ was indeed through the MEK pathway (*Figure 7—figure supplement 1A*). Importantly, the DT-induced RIPK3 was phosphorylated at its serine 165/threonine 166 sites, and the treated cells were undergoing apoptosis, as indicated by the presence of active caspase-3 (*Figure 7C, Figure 7—figure supplement 1B*). Consistent with the notion that apoptosis induced by PGF$_{2\alpha}$ is RIPK3 dependent, the primary granulosa lutein cells isolated from the RIPK3 knockout mice failed to respond to PGF2α activate caspase-3 (*Figure 7C*).

The ability of primary granulosa lutein cells from young animals to undergo RIPK3-mediated apoptosis in response to PGF$_{2\alpha}$ allowed us to genetically verify the role of the RIPK3-RIPK1-FADD-caspase-8 pathway in luteum regression. To this end, we first induced hyper-ovulation in young female mice 4 weeks of age with gonadotropins and then treated them with DT as previously described (*Carambula et al., 2002*; *Carambula et al., 2003*; *Figure 7—figure supplement 1C*). The hyper-ovulation induced by gonadotropins generated multiple corpus lutea in the ovaries of these young mice (*Figure 7D*). The subsequent DT treatment induced the elevation of RIPK3 protein in their corpus lutea, and the induced RIPK3 was phosphorylated at the serine 164/threonine 165 sites (*Figure 7D, E*). Moreover, the treatment of DT caused active caspase-3 signal to appear in the corpus lutea of wild-type ovaries (*Figure 7F, G*). Mice with their *PTGFR* gene knocked out failed to elevate the RIPK3 level in response to PGF$_{2\alpha}$ (*Figure 6—figure supplement 1C, D*, *Figure 7E*). Although DT treatment induced the elevation of phosphorylation-resistant RIPK3(S165A/T166A) protein in the ovaries of $Ripk3^{S165A-T166A/S165A-T166A}$ knock-in mice to the same level as wild-type RIPK3 protein (*Figure 7—figure supplement 1D*), significantly less active caspase-3 signal was observed in these ovaries, similar to the ovaries of $Ripk3^{-/-}$, $Fadd^{-/-}/Mlkl^{-/-}$, or $Ptgfr^{-/-}$ mice, confirming that the observed DT-induced apoptosis in luteal cells was through the RIPK3/FADD-mediated apoptosis pathway triggered by the phosphorylation of serine 165/threonine 166 sites (*Figure 7F, G*). We further co-stained the ovaries with antibodies against active caspase-3 and phospho-S165/T166 RIPK3 and found that most of the positive signals were overlapping (*Figure 7H*), indicating that serine 165/threonine 166 phosphorylated RIPK3 activates apoptosis in corpus luteum, triggered by PGF$_{2\alpha}$. Interestingly, although significantly lower overall, there was more active caspase-3 in $Ripk3^{-/-}$ and

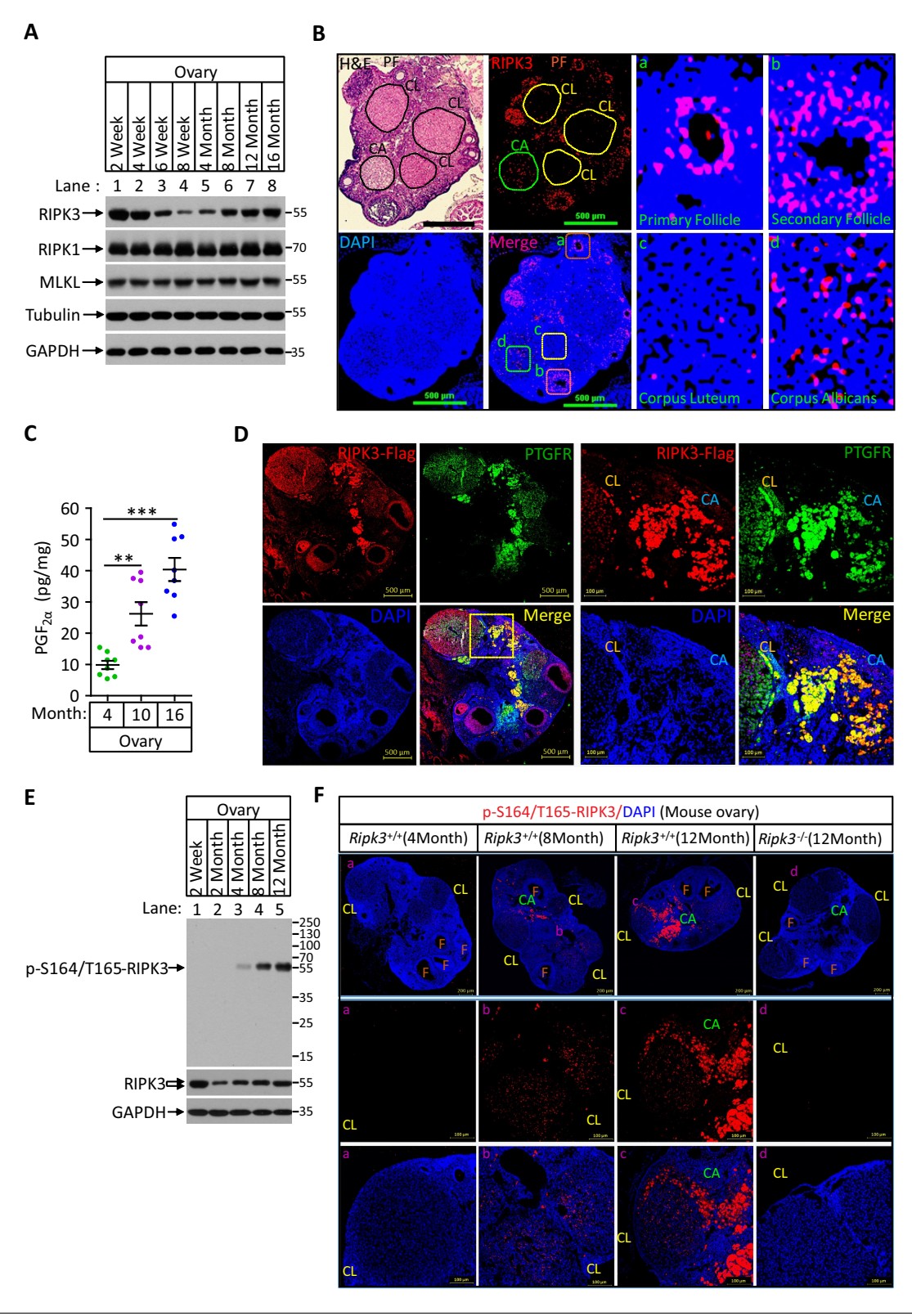

**Figure 6.** Phospho-serine 164/threonine 165-RIPK3 signals were present in aged ovary corpus albican. (**A**) Western blot analysis of RIPK1, RIPK3, and MLKL levels in perfused mouse ovary extracts of different ages. Each group is representative of at least three mice. (**B**) H&E and immunofluorescence (IF) imaging of an 8-month-old ovary. Two adjacent sections were analyzed. One section was stained with H&E, and the other was IF stained with a RIPK3 antibody (red) and DAPI (blue). Scale bar, 500 μm. Higher-power views of selected areas were acquired in a (primordia follicle), b (secondary

*Figure 6 continued on next page*

*Figure 6 continued*

follicle), c (corpus luteum), and d (corpus albicans) as indicated. PF: primary follicle; CL: corpus luteum; CA: corpus albicans. (**C**) Ovarian $PGF_{2\alpha}$ levels of wild-type mice (n = 8) at the indicated age assayed by ELISA. Data represent the mean ± s.e.m. **p<0.01, ***p<0.001. p-values were determined by two-sided unpaired Student's *t*-tests. (**D**) Immunofluorescence images of a RIPK3 C-terminus HA-3xFlag knock-in mouse ovary (n = 5; 12 months) stained with antibodies against prostaglandin F receptor (PTGFR, green) and Flag (red). Counterstaining with DAPI (blue). Scale bar, 500 μm. Higher-power views (right panels) were acquired from the indicated boxed area in the second lower left panel. CL: corpus luteum; CA: corpus albicans. Scale bar, 100 μm. (**E**) Western blot analysis of p-S164/T165-RIPK3 and RIPK3 levels in extracts from perfused ovaries prepared from mice at the indicated age. Each group is representative of at least three mice. (**F**) Immunofluorescence images of ovaries from *Ripk3*$^{+/+}$ and *Ripk3*$^{-/-}$ mice (4 months, 8 months and 12 months; n = 3) at the indicated ages stained with the p-S164/T165-RIPK3 antibody (red). Counterstaining with DAPI (blue). Scale bar, 200 μm. Higher-power views (lower two panels) were acquired from the selected boxed areas from the upper panel. a (CL), b (CL), c (CL, CA), and d (CL). F: follicle; CL: corpus luteum; CA: corpus albicans. Scale bar, 100 μm.

The online version of this article includes the following figure supplement(s) for figure 6:

**Figure supplement 1.** Generation of RIPK3 C-terminus HA-3xFlag knock-in and *Ptgfr*$^{-/-}$ mice.

---

*Ripk3*$^{S165A-T166A/S165A-T166A}$ ovaries than in *Fadd*$^{-/-}$/*Mlkl*$^{-/-}$ ovaries, suggesting that there might be a redundant RIPK3-independent but FADD-dependent apoptosis pathway that also functions in resolving the corpus luteum in this model. This residue cell death could be through the classic extrinsic apoptosis pathway described previously (*Carambula et al., 2003*; *Hu et al., 2001*). Collectively, these results demonstrated that most of the corpus luteum involution induced by $PGF_{2\alpha}$ is through the RIPK3-RIPK1/FADD/caspase-8 pathway triggered by RIPK3 induction and phosphorylation at serine 165/threonine 166.

## Discussion

### The function of RIPK3 in apoptosis and necroptosis is controlled by phosphorylation of the amino acid residue serine 164

The apoptotic role of RIPK3 has been observed ever since RIPK3 was first identified (*Sun et al., 1999*). The initial characterization of RIPK3, mainly by ectopic overexpression, indicated that it functioned in apoptosis (*Sun et al., 1999*). Later, however, RIPK3 was identified as a key signaling molecule for necroptosis due to its ability to specifically phosphorylate MLKL, the executioner of this form of cell death (*Cho et al., 2009*; *He et al., 2009*; *Zhang et al., 2009*; *Sun et al., 2012*). The debate of the apoptotic or necroptotic role of RIPK3 was further brought to the front by the observation that the D161N mutation of RIPK3 (D160N of human origin), a mutation that takes out the key DFG motif of the kinase active site and renders RIPK3 kinase-inactive, caused spontaneous apoptosis by activating caspase-8 through RIPK1 and FADD (*Newton et al., 2014*). Yet surprisingly, another kinase-dead mutant of RIPK3, the K51A mutant, does not induce apoptosis like D161N, but instead only starts to signal to apoptosis in the presence of small-molecule RIPK3 kinase inhibitors that occupy the ATP-binding site of the kinase domain (*Mandal et al., 2014*). These observations suggest that it is not the inactivation of kinase activity per se that switches RIPK3 function from necroptosis to apoptosis. It is more likely the conformational change that happens due to particular mutations or kinase inhibitor binding that triggered such a change. We now propose that this mutation-induced change of RIPK3 function from pro-necroptotic to apoptotic most likely mimicked the specific phosphorylation at the serine 164 (mouse 165) site. The serine 164 amide nitrogen forms a conserved H-bond with the carbonyl oxygen of phenylalanine 160 (part of the DFG motif of kinase), a feature important to maintain the RIPK3 active configuration (*Xie et al., 2013*). Introducing a phosphate group at this site conceivably disrupts this critical interaction, resulting in loss of function of the RIPK3 kinase and a conformational change that allows the RHIM domain to be available for RHIM-RHIM interaction between RIPK3 and RIPK1. Since it is the amide, not the side chain of serine 164 that participates in maintaining the active configuration of RIPK3, it is not surprising that the phosphorylation-resistant mutant S164A/T165A retains the kinase activity and is still able to signal to necroptosis, although such a mutation loses its apoptosis-inducing activity. Interestingly, RIPK1 protein functions here purely as an adaptor for the recruitment of FADD and caspase-8, and its kinase activity is not required for this apoptosis function. Therefore, unlike the apoptosis and necroptosis-inducing functions of RIPK1 when it responds to the activation of the TNF receptor family of

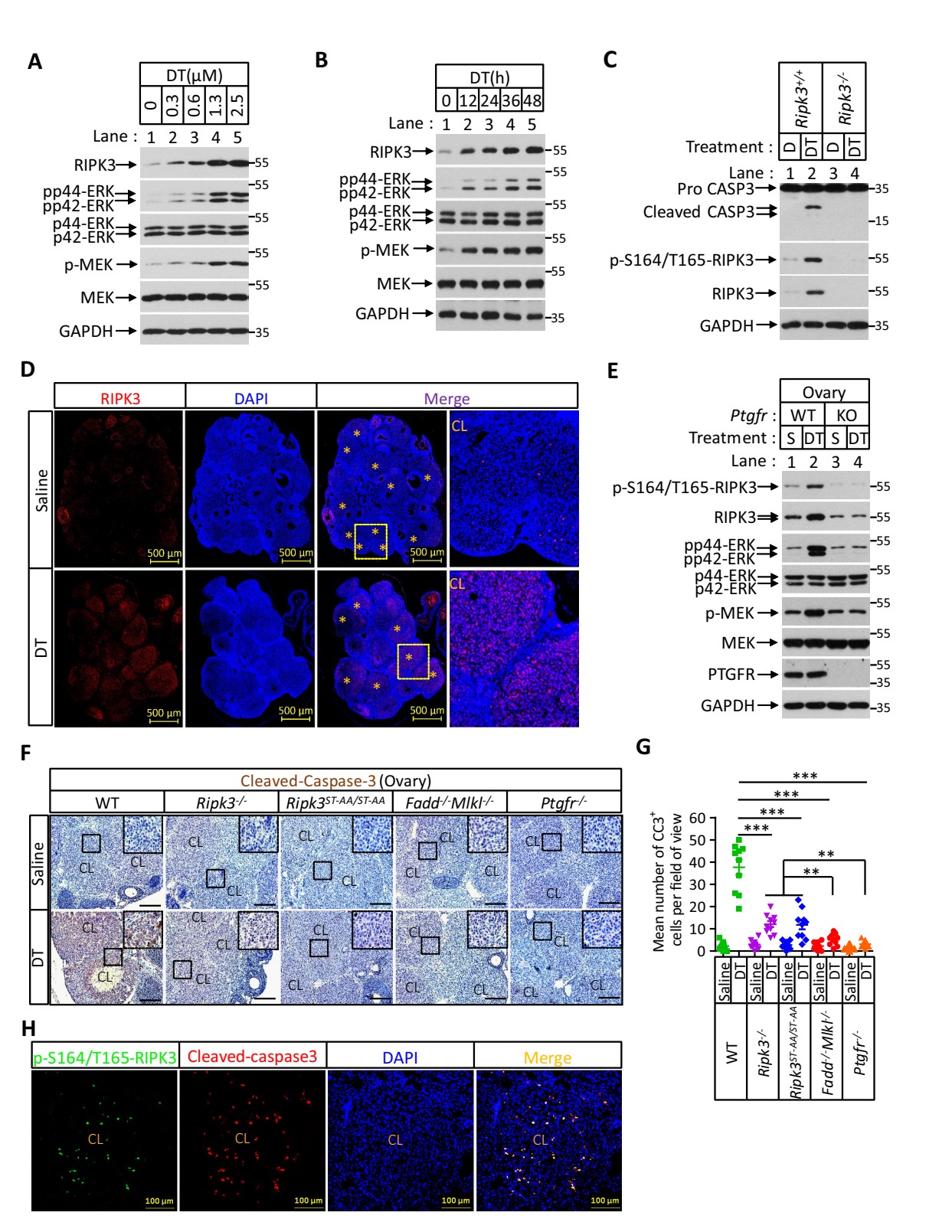

**Figure 7.** Prostaglandin F2alpha (PGF$_{2\alpha}$) induces ovarian RIPK3 expression for corpus luteum involution. (A–C) Primary granulosal lutein cells (WT, *Ripk3*$^{-/-}$) were isolated from 3-month-old mice ovaries. The cells were treated with dinoprost tromethamine (DT) at the indicated concentration for 36 hr in (A); with 1.5 μM DT at the indicated time in (B); or with 1.5 μM DT for 36 hr in (C). The cell lysates from the DT-treated cells were analyzed by western blotting using antibodies as indicated. (D, E) *Ptgfr*$^{+/+}$ and *Ptgfr*$^{-/-}$ littermate female mice (n = 16; 25–26 days) were given 7.5 IU pregnant mare serum

*Figure 7 continued on next page*

*Figure 7 continued*

gonadotropin (PMSG) intraperitoneally (IP) followed by 7.5 IU serum gonadotropin and chorionic gonadotropin (SCG) 46 hr later to synchronize ovulation. The animals were then injected with DT (10 μg, IP) or saline 24 hr post-ovulation. Ovaries were then collected 12 hr later and stained with anti-RIPK3 antibody (red) in (D). The ovary lysates were analyzed by western blotting using antibodies as indicated in (E). * indicates corpus luteum. Counterstaining with DAPI (blue). Scale bar, 500 μm. (F, G) wild-type (WT), *Ripk3*[-/-], *Ripk3*[S165A-T166A/S165A-T166A], *Fadd*[-/-]*Mlkl*[-/-] and *Ptgfr*[-/-] female mice (each group, n = 16; 25–26 days) were treated as in (D, F). Ovaries from each group were then collected 24 hr after injecting with DT and stained with anti-cleaved-caspase-3 antibody in (F). The Cleaved-Caspase3[+] cells were counted in five fields per ovary CL and quantified in (G). Scale bar, 20 μm. Data represent the mean ± s.e.m. **p<0.01, ***p<0.001. p-values were determined by two-sided unpaired Student's *t*-tests. (H) WT female mice (n = 3; 25–26 days) were treated as in (D, F). Ovaries were then collected 12 hr after injecting with DT and stained with anti-cleaved-caspase-3 (red) and p-S164/T165-RIPK3 (green) antibody. Counterstaining with DAPI (blue). Scale bar, 100 μm.

The online version of this article includes the following figure supplement(s) for figure 7:

**Figure supplement 1.** Prostaglandin F2alpha (PGF$_{2\alpha}$) stimulates RIPK3 expression through the MAPK pathway.

proteins, the apoptosis-inducing activity of RIPK1 that is initiated by RIPK3 phosphorylation is not sensitive to RIPK1 kinase inhibitors.

Although we cannot completely rule out the possibility of another yet to be identified kinase(s) that carries out the phosphorylation of serine 164/threonine 165 of RIPK3, our current data favor the idea that the phosphorylation is catalyzed by RIPK3 itself. Both D160N and K50A kinase-dead mutant RIPK3 failed to generate the phospho-serine 164/threonine 165 signal (*Figure 3C, Figure 3— figure supplement 1C*). Especially, the K50 site is not in the proximity of the S164/T165 phosphorylation sites, thus, the loss of phosphorylation is unlikely due to interference with substrate recognition that might have occurred with the D160N mutant, if the phosphorylation is carried out by another kinase.

## The level of RIPK3 chaperone Hsp90/CDC37 determines the cellular necroptotic or apoptotic function of RIPK3

Our investigation has now clearly shown that RIPK3 naturally has dual roles in both necroptosis and apoptosis. In cells in which the RIPK3 kinase predominantly signals to necroptosis, like in human colon cancer HT29 cells, the RIPK3 chaperone Hsp90/CDC37 level is relatively high, which allows RIPK3 to fold into a configuration that can only be activated by upstream signaling molecules, be that either RIPK1 in response to TNF receptor family members, or TRIF and ZBP1/DAI in response to toll-like receptors or Z-RNA, respectively. In this case, RIPK3 kinase activity is silent, and no auto-phosphorylation signal or downstream substrate MLKL phosphorylation occurs before upstream activation signals arrive.

However, in cells in which Hsp90/CDC37 chaperone level is relatively low or artificially inhibited by Hsp90 inhibitors or knocked down by the specific shRNA, the newly generated RIPK3, either by artificial induction (through transient transfection or Dox-induced expression) or by physiological signals like PGF$_{2\alpha}$, can auto-phosphorylate its serine 164/threonine 165 sites of its conserved kinase activation loop, triggering apoptosis through the RIPK3-RIPK1-FADD-caspase-8 pathway. Consistently, mice bearing the phospho-mimic mutations of these sites (S165D/T166E) showed spontaneous activation of apoptosis in all RIPK3-expressing tissues, including lung, intestines, and spleen, and these mice die within a month after birth. However, whether the apoptotic events observed in those S165D/T166E knock-in mouse tissues are caused by the activation of the RIPK1-FADD-caspase-8 pathway still requires genetic verification. if indeed the case, knocking out either *caspase-8* or *FADD* gene in those mice should rescue the phenotype.

We noticed previously that Hsp90 inhibitors developed against human Hsp90 did not block necroptosis in mouse BMDM but was efficacious in BMDM from rat (*Li et al., 2015*). We now know that such a difference is mainly due to species specificity of these Hsp90 inhibitors. Using higher concentrations of human Hsp90 inhibitor 17AAG or knockdown Hsp90 protein in mouse cells, we recapitulated the similar necroptosis to apoptosis switch as in human cells (*Figure 4—figure supplement 1B–E*). We thus realized that the previous conclusion that Hsp90/CDC37 chaperone system is not working in mouse cells is wrongly drawn due to lower efficacy of the Hsp90 inhibitors tested, which was developed to inhibit human Hsp90 (*Li et al., 2015*). The chaperone level of Hsp90/CDC37 relative to RIPK3 is important to dictate the necroptotic or apoptotic function of mouse RIPK3 as well. Consistently, the levels of Hsp90 and CDC37 in the mouse luteum granulosa in which RIPK3-

mediated apoptosis happens are much lower than that in the follicle granulosa (*Figure 5—figure supplement 2*).

Serine 164/threonine 165 phosphorylation of RIPK3 seems to be a unique biomarker for this RIPK3-mediated apoptosis pathway. Our development of a monoclonal antibody that specifically recognizes this phosphorylation event should be useful to identify tissues and cells in which this pathway is activated.

## $PGF_{2\alpha}$ induces RIPK3-mediated apoptosis in luteal granulosa cells for luteum regression

One of the tissues in which this specific anti-phospho-S164/T165 monoclonal antibody picked up a signal was in luteal granulosa cells of aged mouse ovary. The appearance of this phospho-RIPK3 signal in the ovary coincident with the elevation of $PGF_{2\alpha}$ level and indeed, an analog of $PGF_{2\alpha}$ was able to induce the expression of S165/T166 phosphorylated RIPK3 and apoptosis in the primary granulosa cells from ovaries of young wild-type mice (*Figure 7*). The phenomenon of $PGF_{2\alpha}$–induced RIPK3 expression in the mouse ovary granulosa cells allowed us to genetically validate this intracellular apoptotic pathway. In a hyper-ovulation model in which many egg follicles from young mice were induced to mature simultaneously, the accumulated granulosa cells in the ovary of wild-type mice can be triggered into resolution by an agonist (DT) of $PGF_{2\alpha}$ through RIPK3-mediated apoptosis. However, DT could not induce apoptosis and luteum resolution in the ovary of PGF receptor knock-out, RIPK3 knockout, FADD/MLKL double knockout, and $Ripk3^{S165A-T166A/S165A-T166A}$ knock-in mice, confirming this intracellular pathway of apoptosis in vivo.

Although the RIPK3-mediated apoptosis can be recapitulated in this hyper-ovulation model using an agonist (DT) of $PGF_{2\alpha}$ in young mice, the serine 165/threonine 166 phospho-RIPK3 signal only naturally appears in luteal granulosa cells when mice reach a certain age. The RIPK3 knockout and $Ripk3^{S165A-T166A/S165A-T166A}$ knock-in mice do not show any fertility defect in their normal reproductive age, indicating that the $PGF_{2\alpha}$-induced, RIPK3-mediated apoptotic pathway does not have a critical role in luteum resolution during the normal reproductive cycle when the mice are young. It is thus interesting to speculate that such a pathway may have a specific role during mouse ovary aging when all corpora lutea eventually become albicans.

## Materials and methods

**Key resources table**

| Reagent type (species) or resource | Designation | Source or reference | Identifiers | Additional information |
|---|---|---|---|---|
| Cell line (*Homo sapiens*) | HEK293T | ATCC | CRL-11268 | Female |
| Cell line (*Homo sapiens*) | HeLa | ATCC | CCL-2 | Female |
| Cell line (*Homo sapiens*) | HT-29 | ATCC | HTB-38 | Female |
| Cell line (*Homo sapiens*) | MCF7 | ATCC | HTB-22 | Female |
| Cell line (*Homo sapiens*) | KGN | *Nishi et al., 2001* | N/A | Female |
| Cell line (*Homo sapiens*) | L929(*Ripk3⁻/⁻*) | Dr. Xiaodong Wang lab at National Institute of Biological Sciences, Beijing | N/A | |

*Continued on next page*

*Continued*

| Reagent type (species) or resource | Designation | Source or reference | Identifiers | Additional information |
|---|---|---|---|---|
| Cell line (*Homo sapiens*) | KGN/TO-RIPK3 | Dr. Xiaodong Wang lab at National Institute of Biological Sciences, Beijing | N/A | |
| Cell line (*Homo sapiens*) | KGN/TO- RIPK3 (S164A/T165A) | Dr. Xiaodong Wang lab at National Institute of Biological Sciences, Beijing | N/A | |
| Cell line (*Homo sapiens*) | MCF7/TO-RIPK3 | Dr. Xiaodong Wang lab at National Institute of Biological Sciences, Beijing | N/A | |
| Cell line (*Homo sapiens*) | MCF7/TO-RIPK3(D160N) | Dr. Xiaodong Wang lab at National Institute of Biological Sciences, Beijing | N/A | |
| Cell line (*Homo sapiens*) | MCF7/TO-RIPK3(K50A) | Dr. Xiaodong Wang lab at National Institute of Biological Sciences, Beijing | N/A | |
| Cell line (*Homo sapiens*) | MCF7/TO- RIPK3 (S164A/T165A) | Dr. Xiaodong Wang lab at National Institute of Biological Sciences, Beijing | N/A | |
| Cell line (*Homo sapiens*) | MCF7/TO-RIPK3 (*RIPK1$^{-/-}$/Caspase-8$^{-/-}$/ FADD$^{-/-}$/cFLIP$^{-/-}$*) | Dr. Xiaodong Wang lab at National Institute of Biological Sciences, Beijing | N/A | |
| Antibody | Anti-RIPK3 (Rabbit polyclonal) | ProSci | Cat# 2283; RRID:AB_203256 | WB (1:1000) |
| Antibody | Anti-p-S164/T165-RIPK3 (Rabbit monoclonal) | Abcam | Cat# Ab255705; | Firstly described in this paper; WB (1:1000) IF (1:100) |
| Antibody | Anti-RIPK3 (Mouse monoclonal) | LSBio | Cat# LS-C336804 | WB (1:1000) |
| Antibody | Anti-GAPDH-HRP (Mouse monoclonal) | MBL | Cat# M171-1; RRID:AB_10699462 | WB (1:20,000) |

*Continued on next page*

*Continued*

| Reagent type (species) or resource | Designation | Source or reference | Identifiers | Additional information |
|---|---|---|---|---|
| Antibody | Anti-β-actin-HRP (Rabbit polyclonal) | MBL | Cat# PM053-7; RRID:AB_10697035 | WB (1:20,000) |
| Antibody | Anti-Tubulin-HRP (Rabbit polyclonal) | MBL | Cat# PM054-7; RRID:AB_10695326 | WB (1:20,000) |
| Antibody | Anti-Flag-HRP (Mouse monoclonal) | Sigma-Aldrich | Cat# A8592; RRID:AB_439702 | WB (1:10,000) |
| Antibody | Anti-RIPK1 (Rabbit polyclonal) | Cell Signaling | Cat# 3493S; RRID:AB_2305314 | WB (1:1000) |
| Antibody | Anti-cleaved-caspase3 (Rabbit polyclonal) | Cell Signaling | Cat# 9661; RRID:AB_2341188 | WB (1:1000) IF(1:100) |
| Antibody | Anti-Mouse-MLKL (Rabbit polyclonal) | ABGENT | Cat# AP14272b; RRID:AB_11134649 | WB (1:1000) |
| Antibody | Anti-Human-MLKL (Rabbit monoclonal) | Abcam | Cat# ab184718; RRID:AB_2755030 | WB (1:1000) |
| Antibody | Anti-caspase-8 (Mouse monoclonal) | Cell Signaling | Cat# 9746; RRID:AB_2275120 | WB (1:1000) |
| Antibody | anti-FADD (Rabbit polyclonal) | Cell Signaling | Cat# 2782; RRID:AB_2100484 | WB (1:1000) |
| Antibody | Anti-p-S227-RIP3 (Rabbit monoclonal) | Abcam | Cat# ab209384; RRID:AB_2714035 | WB (1:1000) |
| Antibody | Anti-p-S232-RIP3 (Rabbit monoclonal) | Abcam | Cat# ab222302 | WB (1:1000) |
| Antibody | Anti-cFLIP (Rabbit polyclonal) | Abcam | Cat# ab6144; RRID:AB_305314 | WB (1:1000) |
| Antibody | Anti-cleaved-caspase-3 (Mouse monoclonal) | St John's Laboratory | Cat# STJ97448 | IHC (1:100) |
| Antibody | Anti-prostaglandin F2 alpha (PTGFR) (Rabbit polyclonal) | Abcam | Cat# ab203342 | WB (1:1000) IF (1:100) |
| Antibody | Anti-HSP90 (Rabbit polyclonal) | Proteintech | Cat# 13171-1-AP; RRID:AB_2120924 | WB (1:1000) IF (1:100) |
| Antibody | Anti-CDC37 (Rabbit monoclonal) | Abcam | Cat# ab108305; RRID:AB_10861724 | WB (1:1000) IF (1:100) |
| Antibody | anti-FLAG M2 (Mouse monoclonal) | Sigma-Aldrich | Cat# F1840 | IHC (1:100) |
| Antibody | Anti-phospho-p44/ 42 MAPK (Rabbit monoclonal) | Cell Signaling | Cat# 4370S; RRID:AB_2315112 | WB (1:1000) |
| Antibody | Anti-p44/42 MAPK (Rabbit monoclonal) | Cell Signaling | Cat# 4965S | WB (1:1000) |
| Antibody | Anti-phospho-MEK1/2 (Rabbit monoclonal) | Cell Signaling | Cat# 9154S; RRID:AB_2138017 | WB (1:1000) |
| Antibody | Anti-MEK1/2 (Rabbit monoclonal) | Abcam | Cat# ab178876 | WB (1:1000) |
| Antibody | Donkey anti-Mouse, Alexa Fluor 488 (Mouse polyclonal) | Thermo Fisher | Cat# A-21202; RRID:AB_141607 | IF (1:500) |

*Continued on next page*

*Continued*

| Reagent type (species) or resource | Designation | Source or reference | Identifiers | Additional information |
|---|---|---|---|---|
| Antibody | Donkey anti-Mouse, Alexa Fluor 555 (Mouse polyclonal) | Thermo Fisher | Cat# A-31570; RRID:AB_2536180 | IF (1:500) |
| Antibody | Donkey anti-Rabbit, Alexa Fluor 488 (Rabbit polyclonal) | Thermo Fisher | Cat# A-21206; RRID:AB_141708 | IF (1:500) |
| Antibody | Donkey anti-Rabbit, Alexa Fluor 555 (Rabbit polyclonal) | Thermo Fisher | Cat# A-31572; RRID:AB_162543 | IF (1:500) |
| Antibody | Anti-Flag M2 affinity gel | Sigma-Aldrich | A2220 | |
| Recombinant DNA reagent | pWPI-HA-3xFlag-RIPK3 | This paper | N/A | Described in Materials and methods; available upon request |
| Recombinant DNA reagent | pWPI-HA-3xFlag-RIPK3(D160N) | This paper | N/A | Described in Materials and methods; available upon request |
| Recombinant DNA reagent | pWPI-HA-3xFlag-RIPK3(K50A) | This paper | N/A | Described in Materials and methods; available upon request |
| Recombinant DNA reagent | pWPI-HA-3xFlag-RIPK3(AAAA) | This paper | N/A | Described in Materials and methods; available upon request |
| Recombinant DNA reagent | pWPI-HA-3xFlag-mRIPK3 | This paper | N/A | Described in Materials and methods; available upon request |
| Recombinant DNA reagent | pWPI-HA-3xFlag-RIPK3(S164D/T165E) | This paper | N/A | Described in Materials and methods; available upon request |
| Recombinant DNA reagent | pWPI-HA-3xFlag-RIPK3(S164E) | This paper | N/A | Described in Materials and methods; available upon request |
| Recombinant DNA reagent | pWPI-HA-3xFlag-RIPK3(T165E) | This paper | N/A | Described in Materials and methods; available upon request |
| Recombinant DNA reagent | pWPI-HA-3xFlag-RIPK3(S164A) | This paper | N/A | Described in Materials and methods; available upon request |
| Recombinant DNA reagent | pWPI-HA-3xFlag-RIPK3(T165A) | This paper | N/A | Described in Materials and methods; available upon request |
| Recombinant DNA reagent | pWPI-HA-3xFlag-RIPK3(S164A/T165A) | This paper | N/A | Described in Materials and methods; available upon request |
| Recombinant DNA reagent | pWPI-HA-3xFlag-mRIPK3(D161N) | This paper | N/A | Described in Materials and methods; available upon request |
| Recombinant DNA reagent | pWPI-HA-3xFlag-mRIPK3(S165D/T166E) | This paper | N/A | Described in Materials and methods; available upon request |
| Recombinant DNA reagent | pLVX-Tight-HA-3xFlag-RIPK3 | This paper | N/A | Described in Materials and methods; available upon request |
| Recombinant DNA reagent | pLVX-Tight-HA-3xFlag-RIPK3(D160N) | This paper | N/A | Described in Materials and methods; available upon request |

*Continued*

| Reagent type (species) or resource | Designation | Source or reference | Identifiers | Additional information |
|---|---|---|---|---|
| Recombinant DNA reagent | pLVX-Tight-HA-3xFlag-RIPK3(K50A) | This paper | N/A | Described in Materials and methods; available upon request |
| Recombinant DNA reagent | pLVX-Tight-HA-3xFlag-RIPK3(S164A/T165A) | This paper | N/A | Described in Materials and methods; available upon request |
| Recombinant DNA reagent | pX458-GFP-RIPK1 | This paper | N/A | Described in Materials and methods; available upon request |
| Recombinant DNA reagent | pX458-GFP-caspase-8 | This paper | N/A | Described in Materials and methods; available upon request |
| Recombinant DNA reagent | pX458-GFP-FADD | This paper | N/A | Described in Materials and methods; available upon request |
| Recombinant DNA reagent | pX458-GFP-cFLIP | This paper | N/A | Described in Materials and methods; available upon request |
| Peptide, recombinant protein | 3xFlag peptide | ChinaPeptides | DYKDHDGDYKDHDIDYKDDDDK | 1 mg/ml |
| Software, algorithm | ImageJ | NIH | N/A | |
| Software, algorithm | Photoshop | Adobe | N/A | |
| Software, algorithm | Lasergene | DNASTAR | N/A | |
| Software, algorithm | GraphPad | GraphPad Software | N/A | |
| Software, algorithm | Prism | GraphPad Software | N/A | |
| Software, algorithm | Nikon A1-R | Nikon | https://www.nikoninstruments.com/Products/Confocal-Microscopes/A1R-HD | |

## Animals

The *Ripk3*[-/-] and *Mlkl*[-/-] mice (C57BL/6J strain) have been described previously (*He et al., 2009*; *Ying et al., 2018*). *Fadd*[+/-]*Mlkl*[-/-] mice were provided by Dr. Haibing Zhang (Institute for Nutritional Sciences, Shanghai, China) (±). C57BL/6J wild-type (WT) mice were obtained from Vital River Laboratory Co. WT, *Ripk3*[-/-], *Mlkl*[-/-], and *Fadd*[-/-]*Mlkl*[-/-] mice were produced and maintained at the SPF animal facility of the National Institute of Biological Sciences, Beijing. *Ripk3*[-/-] mice were produced by mating *Ripk3*[-/-] males with *Ripk3*[-/-] females. *Fadd*[-/-]*Mlkl*[-/-] mice were produced by mating *Fadd*[+/-]-*Mlkl*[-/-] males with *Fadd*[+/-] ±l[-/-] females. *Ripk3*[S165D-T166E/ S165D-T166E] and *Ripk3*[S165A-T166A/ S165A-T166A] knock-in mice were generated using the CRISPR-Cas9 system (*Figure 5—figure supplement 1*). The *Ripk3* C terminus HA-3xFlag knock-in and *Ptgfr* knockout mice were generated using the CRISPR-Cas9 system (*Figure 6—figure supplement 1*).

The primers used for genotyping are listed below:

*Ripk3*-(WT)-F: 5′-GCTAGCAGATTTTGGCCTGTCCACG-3′;
*Ripk3*-(S165D/T166E)-F: 5′-GCTAGCAGATTTTGGCCTGGACGAA-3′;
*Ripk3*-(S165A/T166A)-F: 5′-GCTAGCAGATTTTGGCCTGGCAGCA-3′;
*Ripk3*-R: 5′-GGCCTCTGGCGAGACTTCTTTCCTG-3′;
*Ptgfr*-WT-F: 5′-GCTGTGTTAGCCCATTGAGTCAGGTAGA-3′;
*Ptgfr*-KO-F: 5′-TGATGGTGTCAGTTTGGGCGGTAT-3′;

*Ptgfr*-KO-R: 5′-GCTTTACTTCTGCTACTGAATTCCCTTGG-3′

## Mouse husbandry

Mice were housed in a 12 hr light/dark (light between 08:00 and 20:00) in a temperature-controlled room (21.1 ± 1℃) at the National Institute of Biological Sciences with free access to water. The ages of mice are indicated in the figure, figure legends, or methods. All animal experiments were conducted following the Ministry of Health national guidelines for the housing and care of laboratory animals and were performed in accordance with institutional regulations after review and approval by the Institutional Animal Care and Use Committee at the National Institute of Biological Sciences, Beijing.

## Cell lines and cell cultures

All cells were cultured at 37℃ with 5% $CO_2$. All cell lines were cultured as follows: HT29 cells were obtained from ATCC and cultured in McCoy's 5A culture medium (Invitrogen). HEK293T, HeLa, and MCF7 cells were obtained from ATCC and cultured in DMEM (Hyclone). L929(*Ripk3*$^{-/-}$) cells were cultured in DMEM (Hyclone) (*Ying et al., 2018*). Human granulosa tumor cell line (KGN) (*Nishi et al., 2001*) were kindly provided by Dr. Qiao Jie from Peking University Third Hospital in China and were cultured in DMEM:F12 Medium (Hyclone). Stable KGN or MCF7 cell lines expressing the Tet repressor (KGN-Tet-On cells and MCF7-Tet-On cells) were selected with 1 mg/ml G418 after being infected with virus encoding Tet repressor. KGN-Tet-On and MCF7-Tet-On cells were infected with virus encoding 3xFlag-RIPK3 or another mutant and were selected with 10 μg/ml puromycin to establish the KGN/TO-RIPK3/RIPK3(S164A/T165A) and MCF7/TO-RIPK3/RIPK3(D160N)/RIPK3 (S164A/T165A) cell lines. All media were supplemented with 10% FBS (Thermo Fisher) and 100 units/ml penicillin/streptomycin (Thermo Fisher).

## Plasmids constructs

Full-length human RIPK3, RIPK3(AAAA) (RIPK1 interaction mutant RIPK3, the residues 459–462 within RHIM region were mutated to four alanine residues) and mouse RIPK3 cDNA were kept in our lab and subcloned into the pWPI vector (GFP-tagged) to generate pWPI-HA-3xFlag-RIPK3/RIPK3 (D160N)/RIPK3(AAAA) and pWPI-HA-3xFlag-mRIPK3 virus construct. Using Quickchange Site-Directed Mutagenesis Kit to generate pWPI-HA-3xFlag-RIPK3(D160N)/RIPK3(S164D/T165E)/RIPK3 (S164E)/RIPK3(T165E)/RIPK3(S164A/T165A)/RIPK3(S164A)/RIPK3(T165A) and pWPI-HA-3xFlag-mRIPK3(D161N)/mRIPK3(S165D/T166E) virus construct.pLVX-Tight-Puro Vector and pLVX-Tet-On Advanced were kindly provided by Dr. Wenhui Li lab at NIBS. Full-length human RIPK3 or RIPK3 mutant were subcloned into the pLVX-Tight-Puro Vector to generate pLVX-Tight-HA-3xFlag-RIPK3/ RIPK3(D160N)/RIPK3(S164A/T165A) construct. The gRNAs for targeting *RIPK1* (5′-ATGCTCTTAC-CAGGAAATGT-3′), *Caspase-8* (5′-TGATCGACCCTCCGCCAGAA-3′), *FADD* (5′-AGTCG TCGACGCGCCGCAGC-3′), and *cFLIP* (5′-TACCAGACTGCTTGTACTTC-3′) were designed and cloned into the gRNA-Cas9 expression plasmid pX458-GFP to generate pX458-GFP-RIPK1/caspase-8/FADD/cFLIP construct.

## Cell survival assay

Cell survival assay was performed using Cell Titer-Glo Luminescent Cell Viability Assay kit. A Cell Titer-Glo assay (Promega, G7570) was performed according to the manufacturer's instructions. Luminescence was recorded with a Tecan GENios Pro plate reader.

## Transfections

HeLa or HEK293T cells were transfected with plasmids using Lipofectamine 3000 (Thermo Fisher Scientific) following the manufacturer's instructions.

## Virus packaging

To prepare the virus, HEK293T cells in the 10 cm dish were transfected with 15 μg of pWPI Vector (Vec, control viruses), pWPI-HA-3xFlag-RIPK3/RIPK3(D160N)/RIPK3(AAAA)/RIPK3(S164D/T165E)/ RIPK3(S164E)/RIPK3(T165E)/RIPK3(S164A/T165A)/RIPK3(S164A)/RIPK3(T165A), pLVX-Tight-3xFlag-RIPK3/RIPK3(D160N)/RIPK3(S164A/T165A) or pLVX-Tet-On Advanced DNA together with 11.25 μg

of psPAX2 and 3.75 µg of pMD2.G. 8 hr after transfection, the media were changed to high-serum DMEM (20% FBS with 25 mM HEPES). Another 40 hr later, the media were collected and centrifuged at 3000 rpm for 10 min. The supernatant was filtered through a 0.22 µm membrane, and aliquots of 15 ml were stored at −80℃.

## CRISPR/Cas9 knockout cells

4 µg of pX458-GFP-RIPK1/caspase-8/FADD/cFLIP plasmid was transfected into $1 \times 10^7$ MCF7/TO-RIPK3 cells using the Transfection Reagent (FuGENEHD) by following the manufacturer's instructions. 3 days after the transfection, GFP-positive live cells were sorted into single clones by using a BD FACSArial cell sorter. The single clones were cultured into 96-well plates for another 10–14 days or longer, depending upon the cell growth rate. The anti-RIPK1/caspase-8/FADD/cFLIP immunoblotting was used to screen for the MCF7/TO-RIPK3 ($RIPK1^{-/-}/Caspase-8^{-/-}/FADD^{-/-}/cFLIP^{-/-}$) clones. Genome type of the knockout cells was determined by DNA sequencing.

## Western blotting

Western blotting was performed as previously described. Cell pellet samples were collected and re-suspended in lysis buffer (100 mM Tris-HCl, pH 7.4, 100 mM NaCl, 10% glycerol, 1% Triton X-100, 2 mM EDTA, Roche complete protease inhibitor set, and Sigma phosphatase inhibitor set), incubated on ice for 30 min, and centrifuged at $20,000 \times g$ for 30 min. The supernatants were collected for western blotting. Ovary or other tissue were ground and re-suspended in lysis buffer, homogenized for 30 s with a Paddle Blender (Prima, PB100), incubated on ice for 30 min, and centrifuged at $20,000 \times g$ for 30 min. The supernatants were collected for western blotting.

## Immunoprecipitation

The cells were cultured on 10 cm dishes and grown to confluence. Cells at 70% confluence with or without Dox (1 µg/ml) induction for 24 hr. Then cells were washed once with PBS and harvested by scraping and centrifugation at $800 \times g$ for 5 min. The harvested cells were washed with PBS and lysed for 30 min on ice in the lysis buffer. Cell lysates were then spun down at $12,000 \times g$ for 20 min. The soluble fraction was collected, and the protein concentration was determined by Bradford assay. Cell extracted was mixed with anti-Flag affinity gel (Sigma-Aldrich, A2220) in a ratio of 1 mg of extract per 30 µl of agarose. After overnight rocking at 4℃, the beads were washed three times with lysis buffer. The beads were then eluted with 0.5 mg/ml of the corresponding antigenic peptide for 6 hr or directly boiled in $1 \times$ SDS loading buffer (125 mM Tris, PH 6.8, 2% 2-mercaptoethanol, 3% SDS, 10% glycerol, and 0.01% bromophenol blue).

## Tissue collection

Mice were sacrificed and perfused with PBS. Tissues for stain comparisons were taken at the same stage of the estrous cycle: ovarian were taken from all mice during diestrus. Major organs were removed, cut into appropriately sized pieces, and either flash-frozen in liquid nitrogen and stored at −80℃ or placed in 4% paraformaldehyde for preservation. After several days of 4% paraformaldehyde fixation at room temperature, tissue fragments were transferred to 70% ethanol and stored at 4℃.

## Histology and immunohistochemistry

Paraffin-embedded specimens were sectioned to a 5 µm thickness and were then deparaffinized, rehydrated, and stained with hematoxylin and eosin (H&E) using standard protocols. For the preparation of the immunohistochemistry samples, sections were dewaxed, incubated in boiling citrate buffer solution for 15 min in plastic dishes, and subsequently allowed to cool down to room temperature over 3 hr. Endogenous peroxidase activity was blocked by immersing the slides in hydrogen peroxide buffer (10%, Sinopharm Chemical Reagent) for 15 min at room temperature and were then washed with PBS. Blocking buffer (1% bovine serum albumin in PBS) was added, and the slides were incubated for 2 hr at room temperature. Primary antibody against Flag, RIPK3, cleaved-caspase-3, and p-S164/T165-RIPK3 was incubated overnight at 4℃ in PBS. After three washes with PBS, slides were incubated with secondary antibody (polymer-horseradish-peroxidase-labeled anti-rabbit, Sigma) in PBS. After a further three washes, slides were analyzed using a diaminobutyric acid

substrate kit (Thermo Fisher). Slides were counterstained with hematoxylin and mounted in neutral balsam medium (Sinopharm Chemical).

Immunohistochemistry analysis for RIPK3 or p-S164/T165-RIPK3 was performed using an antibody against RIPK3 and p-S164/T165-RIPK3. Primary antibody against cleaved-caspase-3 or p-S164/T165-RIPK3 was incubated overnight at 4°C in PBS. After three washes with PBS, slides were incubated with DyLight-555/488 conjugated donkey anti-rabbit/mouse secondary antibodies (Thermo Fisher) in PBS for 8 hr at 4°C. After a further three washes, slides were incubated with RIPK3 or p-S164/T165-RIPK3 antibody overnight at 4°C in PBS. After a further three washes, slides were incubated with DyLight-488/555 conjugated donkey anti-mouse/rabbit secondary antibodies (Thermo Fisher) for 2 hr at room temperature in PBS. After a further three washes in PBS, the cell nuclei were then counterstained with DAPI (Thermo Fisher) in PBS. Fluorescence microscopy was performed using a Nikon A1-R confocal microscope.

## Isolation of the bone marrow-derived macrophages

Macrophages were isolated from the $Ripk3^{+/+}$ and $Ripk3^{S165D-T166E/S165D-T166E}$ littermate mice and a caspase inhibitor z-VAD-fmk was added when culture these BMDMs. We noticed that the caspase inhibitor was needed to prevent spontaneous BMDM cell death when isolated from $Ripk3^{S165D-T166E/S165D-T166E}$ mice.

## Isolation of mice granulosal lutein cells from ovaries (*Newton et al., 1999*; *Tian et al., 2015*)

Ovaries from 3-month-old wild-type mice were collected and placed in Enriched DMEM:F12 (Hyclone) media and placed on ice. Ovaries were ground on the strainer (40 μm) with forceps. Centrifuge the cell suspension at 900 × g for 5 min at 4°C, wash three times with PBS. The cells pellet were incubated with trypsin for 15–20 min at 37°C in a shaking water bath at 80 oscillations (osc)/min and were then layered over 40 ml 5% Percoll/95% 1× Hank's balanced salt solution in a 50 ml conical tube and allowed to settle for 20 min. After incubation, 3 ml charcoal-stripped FBS was immediately added to halt the digestion. All fractions were mixed and immediately centrifuged at 900 × g for 10 min at 4°C. Pellets were re-suspended in PBS and washed three times, then cultured in DMEM:F12 (10% FBS) medium at 37°C with 5% $CO_2$. After 24 hr, the suspended granulosal lutein cells were collected and cultured in a new cell dish. The adherent follicular granulosa cells were then treated with the indicated stimuli. The cell lysates were analyzed by western blotting.

## In vivo studies of luteal regression

Wild-type, $Ripk3^{-/-}$, $Fadd^{-/-}Mlkl^{-/-}$, $Ripk3^{S165A-T166A/ S165A-T166A}$, and $Ptgfr^{-/-}$ female mice (each group, n = 16) at day 25–26 postpartum were intraperitoneally injected with 7.5 IU (100 μl/mouse) pregnant mare serum gonadotropin (PMSG) followed by intraperitoneal injection with 7.5 IU (100 μl/mouse) serum gonadotropin and chorionic gonadotropin (SCG) 46 hr later to induce superovulation. Those mice were injected with DT (10 μg, IP) or saline at 24 hr after post-ovulation. Ovaries were then collected at 12 or 24 hr post-injection. In some experiments, ovaries were ground and re-suspended in lysis buffer, homogenized for 30 s with a Paddle Blender (Prima, PB100), incubated on ice for 30 min, and centrifuged at 20,000 × g for 30 min. The supernatants were analyzed by western blotting. In other experiments, ovaries were fixed with 4% paraformaldehyde, paraffin-embedded, serially sectioned (5 μm), and mounted in order on glass microscope slides. Ovarian mid-sagittal sections were used for immunohistochemical analysis with RIPK3 or cleaved-caspase-3 antibody.

## Mass spectrometry and data analysis

HeLa/TO-RIPK3 and MCF7/TO-RIPK3 cells were treated with Dox plus z-VAD for 24 hr. Then the cell extracts were prepared and used for immunoprecipitation with an anti-Flag antibody. The immunoprecipitates were washed three times with lysis buffer. The beads were then eluted with 0.5 mg/ml of the corresponding antigenic peptide for 6 hr or directly boiled in 1× SDS loading buffer and subjected to SDS-PAGE. RIPK3 bands were excised from SDS-PAGE gel and then dissolved in 2 M urea, 50 mM ammonium bicarbonate, pH 8.0, and reduced in 2 mM DTT at 56°C for 30 min followed by alkylation in 10 mM iodoacetamide at dark for 1 hr. Then the protein was digested with sequencing grade modified trypsin (Promega) (1: 40 enzyme to total protein) at 37°C overnight. The tryptic

peptides were separated by an analytical capillary column (50 μm × 15 cm) packed with 5 μm spherical C18 reversed phase material (YMC, Kyoyo, Japan). A Waters nanoAcquity UPLC system (Waters, Milford, USA) was used to generate the following HPLC gradient: 0–30% B in 40 min, 30–70% B in 15 min (A = 0.1% formic acid in water, B = 0.1% formic acid in acetonitrile). The eluted peptides were sprayed into a LTQ Orbitrap Velos mass spectrometer (Thermo Fisher Scientific, San Jose, CA, USA) equipped with a nano-ESI ion source. The mass spectrometer was operated in data-dependent mode with one MS scan followed by four collision-nduced dissociation and four high-energy collisional dissociation MS/MS scans for each cycle. Database searches were performed on an in-house Mascot server (Matrix Science Ltd, London, UK) against Human RIPK3 protein sequence. The search parameters are 7 ppm mass tolerance for precursor ions; 0.5 Da mass tolerance for product ions; three missed cleavage sites were allowed for trypsin digestion and the following variable modifications were included: oxidation on methionine, cysteine carbamidomethylation, and serine, threonine, and tyrosine phosphorylation.

### Statistical analysis

All experiments were repeated at least twice, with similar results. Data represent biological replicates. Statistical tests were used for every type of analysis. The data meet the assumptions of the statistical tests described for each figure. Results are expressed as the mean ± s.e.m or SD. Differences between experimental groups were assessed for significance using a two-sided unpaired Student's $t$-test using GraphPad prism5 and Excel software. The $*p<0.05$, $**p<0.01$, and $***p<0.001$ levels were considered significant. NS indicates not significant.

## Acknowledgements

We thank Dr. Alex Wang for critically reading and editing the manuscript. This work was supported by institutional grants from the Chinese Ministry of Science and Technology and Beijing Municipal Commission of Science and Technology. The funders had no role in study design, data collection and interpretation, or the decision to submit the work for publication.

## Additional information

### Funding

| Funder | Grant reference number | Author |
| --- | --- | --- |
| Ministry of Science and Technology of the People's Republic of China | Institutional | Xiaodong Wang |

The funders had no role in study design, data collection and interpretation, or the decision to submit the work for publication.

### Author contributions

Dianrong Li, Conceptualization, Data curation, Formal analysis, Validation, Investigation, Methodology, Project administration, Writing - review and editing; Jie Chen, Data curation, Formal analysis, Investigation; Jia Guo, Resources, Investigation; Lin Li, Gaihong Cai, She Chen, Investigation; Jia Huang, Hui Yang, Yinhua Zhuang, Fengchao Wang, Resources; Xiaodong Wang, Conceptualization, Data curation, Funding acquisition, Writing - original draft, Writing - review and editing

### Author ORCIDs

Dianrong Li ⓘ https://orcid.org/0000-0002-5564-3033
Xiaodong Wang ⓘ https://orcid.org/0000-0001-9885-356X

### Ethics

Animal experimentation: This study was performed in strict accordance with the recommendations in the Guide for the Care and Use of Laboratory Animals of the National Institutes of Biological Sciences, Beijing (Approval ID: NIBSLuoM15C).

Decision letter and Author response
Decision letter https://doi.org/10.7554/eLife.67409.sa1
Author response https://doi.org/10.7554/eLife.67409.sa2

## Additional files

### Supplementary files
• Transparent reporting form

### Data availability
All data generated or analysed during this study are included in the manuscript and figure supplements.

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
