## [Decision Letter]

**Acceptance summary:**

The protein kinase RIPK3 is widely known to promote a form of lytic cell death termed necroptosis. However, RIPK3 can also promote apoptotic cell death under certain conditions. However, the mechanism by which RIPK3 promotes apoptosis and the physiological relevance of this apoptotic activity were not understood. The study addresses this gap and identifies that phosphorylation of RIPK3 on Ser164 and Thr165 appears to inhibit RIPK3's capacity to induce necroptosis and make it a potent inducer of apoptosis. Such RIPK3 phosphorylation is found to regulate corpus luteum regression. This paper is likely to be of interest to the cell death, protein phosphorylation, and reproductive biology communities.

**Decision letter after peer review:**

Thank you for submitting your article "A phosphorylation of RIPK3 kinase initiates an intracellular apoptotic pathway that promotes corpus luteum regression" for consideration by *eLife*. Your article has been reviewed by 2 peer reviewers, and the evaluation has been overseen by a Reviewing Editor and Philip Cole as the Senior Editor. The reviewers have opted to remain anonymous.

Summary:

The protein kinase RIPK3 was widely known to promote a form of lytic cell death termed necroptosis. However, RIPK3 could also promote apoptotic cell death under certain conditions. However, the mechanism by which RIPK3 promotes apoptosis and the physiological relevance of this apoptotic activity was not understood. The study aims to address this gap.

The study identifies a new phosphorylation event on RIPK3 (S164/T165) that appears to inhibit its capacity to induce necroptosis and make it a potent inducer of apoptosis. Low levels of the chaperone HSP90/CDC37 are posited to favor S164/T165 RIPK3 phosphorylation. RIPK3 activity identified in this work is positioned in the context of luteal regression by inducing apoptosis in luteal granulosa cells in the ovaries of female mice. Outcomes of in vitro mechanistic studies, based on overexpression of the different mutants in cell lines would be interesting to extend to cells from the knock-in mice expressing the mutated proteins at endogenous levels. Further, characterization of phenotypes of mice that lack RIPK3 kinase activity vis-a-vis mice that harbor mutations that mimic this phosphorylation would further strengthen the conclusions of this study.

Essential revisions:

The authors showed that doxycycline-induced RIPK3 in the human cancer cells MCF7 and KGN led to apoptosis and phosphorylation at S164/T165. However, similar Dox-induced RIPK3 expression in HeLa cells did not promote S164/T165 phosphorylation or apoptosis. The authors attributed this differential response to the lower expression of the RIPK3 chaperone HSP90/Cdc37 in MCF7 and KGN cells. The phosphomimetic mutant S164D/T165E RIPK3 was defective for classical TZB-induced necroptosis but induced apoptosis in HeLa cells despite high HSP90/CDC37 expression. By contrast, S164A/T165A-RIPK3 failed to cause apoptosis but restored TBZ-induced necroptosis in MCF7 and KGN cells. The authors propose that this process is particularly important in luteal granulosa cells and provide evidence suggesting that RIPK3 phosphorylation on S164/T165 occurs in the ovaries of older mice. This seems counterintuitive given that corpus luteum involution occurs as part of the ovulation cycle and should therefore be especially relevant in young, sexually mature mice.

The authors are commended for the amount of work they have put in this study, going from the identification of this phosphorylation site to in vitro and in vivo functional studies including the generation of knock-in mice to address its function. The authors have successfully identified a protein phosphorylation that controls the form of cell death mediated by RIPK3. Key questions remain as to how this modification prevents RIPK3 from promoting necroptosis. There are some aspects of the study that could be strengthened, particularly regarding the mechanism of regulation and the physiological significance of this phosphorylation event.

1. Using cells from the knock-in mice expressing the mutated proteins at endogenous levels would be a more appropriate experimental system to assess the mechanistic underpinnings such as the interaction with HSP90/CDC37. For example, in figure 5G, cell death can be assessed in BMDMs from both the phosphomimetic and the alanine mutants in a more comprehensive manner. Do BMDMs or other primary cells (MEFs, lymphocytes, etc) from the phosphomimetic mutant knock-in mice undergo spontaneous apoptosis in the absence of zVAD, and if not why?

2. The authors implied that the kinase activity of RIPK3 is critical in this switch to apoptosis. However, the phenotypes of mice that lack RIPK3 kinase activity do not match that of the mice that harbor mutations that mimic this phosphorylation. Do RIPK3 deficient or S165a/T166A mutant mice show expected reproductive defects that may be attributed to the lack of the proposed RIPK3-mediated apoptosis program in luteal granulosa cells?

3. The heterozygous expression of the phosphomimetic mutants does not cause any pathology in vivo. The authors speculate that a threshold of expression is required for activation of this mutant, however, an alternative explanation could be that the presence of the wild-type protein prevents its activation, e.g. by trans-autophosphorylation on S227. Introducing a RIPK3 null allele to generate heterozygous RIPK3S165D/T166E mice that do not express wild-type RIPK3 could in principle help resolve this question, as in that case the phosphomimetic mutant will be expressed at the same level but in the absence of the wild type protein. We hope the authors can discuss this further.

4. The in vivo data in the knock-in mouse models clearly show that phosphomimetic mutations (RIPK3S165D/T166E) on RIPK3 cause severe pathology in multiple organs associated with increased numbers of dying cells. However, definitive evidence that the pathology is induced by apoptosis is not provided. The authors should incorporate a statement that clarifies this or provide evidence implicating an apoptotic pathway. For instance, rescue experiments by crossing to caspase-8 knockout mice would be one mechanism to establish that pathology is indeed induced by apoptosis.

5. The histological data and particularly the cleaved caspase-3 immunostainings are very hard to evaluate and could be improved. A question that deserves some discussion in the paper is also why the S165D/T166E mice don't die during embryonic life like the D161N mutants.

6. Information on the reproducibility of the experiments, including technical versus biological replicates and how many times each experiment was repeated, is not included in many figures. In addition, for all in vitro experiments in transfected and CRISPR knockout clones, it would help to include information on the number of independent clones tested. Statistical analysis needs to be included in all figures.

7. RIPK3 immunoblots often show 2 or three bands, which sometimes seem to correspond to phosphorylated versions and other times FLAG-tagged versus wild type. It would be helpful to label these accordingly in the figures to make the data easier for the readers to assess.

*Reviewer #1:*

The protein kinase RIPK3 was widely known to promote a form of lytic cell death termed necroptosis. However, RIPK3 could also promote apoptotic cell death under certain conditions. However, the mechanism by which RIPK3 promotes apoptosis and the physiological relevance of this apoptotic activity were not understood. In this study, the authors provided answers to these two questions.

Strengths:

The authors found that a specific phosphorylation on RIPK3 plays a critical role in the switch of RIPK3 into an apoptosis-inducing protein. The authors provided strong evidence to support their conclusion using mouse genetics and demonstrated a role for this RIPK3 activity in reproductive physiology.

Weaknesses:

Although the authors succeeded in finding the protein phosphorylation that controls the form of cell death mediated by RIPK3, key questions remained as to how this modification prevents RIPK3 from promoting necroptosis. Also, the authors implied that the kinase activity of RIPK3 is critical in this switch to apoptosis. However, the phenotypes of mice that lack RIPK3 kinase activity do not match that of the mice that harbor mutations that mimic this phosphorylation.

Overall, this work should provide useful information for future studies to further examine the mechanism by which RIPK3 controls different types of cell death in normal and pathophysiology.

*Reviewer #2:*

The authors sought to understand the mechanisms determining whether the kinase RIPK3 induces apoptosis or necroptosis and the physiological significance of this dual function. They identified a new phosphorylation event on RIPK3 (S164/T165) that appears to inhibit its capacity to induce necroptosis and make it a potent inducer or apoptosis. Low levels of the chaperone HSP90/CDC37 seem to favor S164/T165 RIPK3 phosphorylation, which is suggested to be important for luteal regression by inducing apoptosis in luteal granulosa cells in the ovaries of female mice.

The results presented expand on previous studies showing that whereas RIPK3 induces necroptosis by phosphorylating MLKL, inhibition of RIPK3 kinase activity by small molecules or by D160N mutation caused apoptosis and embryonic lethality. The authors provide experimental evidence supporting that phosphorylation on S164/T165 promotes apoptosis in vitro and in vivo, however the mechanisms regulating this transition remain poorly understood. The data on HSP90/CDC37 is supportive but largely correlative. The authors speculate that association with this chaperone is necessary for proper folding of RIPK3 into a configuration that can only be activated by upstream necroptosis inducers, while at low HSP90/CDC37 levels RIPK3 is not correctly folded and likely auto-phosphorylates on S164/T165, however this remains to be demonstrated. The authors propose that this process is particularly important in luteal granulosa cells and provide some evidence suggesting that RIPK3 phosphorylation on S164/T165 occurs in the ovaries of older mice. This seems counterintuitive given that corpus luteum involution occurs as part of the ovulation cycle and should therefore be especially relevant in young, sexually mature mice. Most importantly, there is no evidence that RIPK3 phosphorylation at these sites is important for female reproductive function, questioning its physiological significance. It would be important to know whether RIPK3 deficient or S165a/T166A mutant mice show any reproductive defects that would be expected by the lack of the proposed RIPK3-mediated apoptosis program in luteal granulosa cells.

The in vivo data in the knock-in mouse models clearly show that phosphomimetic mutations (RIPK3S165D/T166E) on RIPK3 cause severe pathology in multiple organs associated with increased numbers of dying cells. However, rescue experiments, for example by crossing to caspase-8 knockout mice, to prove that the pathology is indeed induced by apoptosis are lacking. It is also interesting that heterozygous expression of the phosphomimetic mutants does not cause any pathology in vivo. The authors speculate that a threshold of expression is required for activation of this mutant, however an alternative explanation could be that the presence of the wild type protein prevents its activation, e.g. by trans-autophosphorylation on S227. Introducing a RIPK3 null allele to generate heterozygous RIPK3S165D/T166E mice that do not express wild type RIPK3 could help resolve this question, as in that case the phosphomimetic mutant will be expressed at the same level but in the absence of the wild type protein.

Finally, most of the in vitro mechanistic studies rely on overexpression of the different mutants in cell lines. Using cells from the knock-in mice expressing the mutated proteins at endogenous levels would be a more appropriate experimental system to explore the mechanistic underpinnings such as the interaction with HSP90/CDC37.

[Editors' note: further revisions were suggested prior to acceptance, as described below.]

Thank you for submitting your article A phosphorylation of RIPK3 kinase initiates an intracellular apoptotic pathway that promotes prostaglandin_2α_-induced corpus luteum regression" for consideration by *eLife*. Your article has been reviewed by a Reviewing Editor and Philip Cole as the Senior Editor. No reviewers found for this submission.

The authors have largely been responsive to the prior concerns raised and the paper needs to deal with one more issue prior to acceptance.

Summary:

The protein kinase RIPK3 is widely known to promote a form of lytic cell death termed necroptosis. RIPK3 is also reported to promote apoptotic cell death under certain conditions. However, the mechanism by which RIPK3 promotes apoptosis and the physiological relevance of this apoptotic activity was not understood. This study identifies a new phosphorylation event on RIPK3 (S164/T165) that appears to inhibit its capacity to induce necroptosis and make it a potent inducer or apoptosis. RIPK3 activity identified in this work is positioned in the context of luteal regression by inducing apoptosis in luteal granulosa cells in the ovaries of female mice. The study opens up new questions about mechanisms underpinning these outcomes, including apoptotic pathways activated in this context.

Essential revisions:

1. Since the S165/T166 phosphorylation mutant does not fully phenocopy that of the RIPK3 kinase-inactive mutants, the possibility that the underlying mechanisms by which these mutants promote apoptosis (i.e. different phenotypes of the respective mutant mice) is not ruled out. The authors should clarify this further in the text.

---

## [Author Response]

Essential revisions:The authors showed that doxycycline-induced RIPK3 in the human cancer cells MCF7 and KGN led to apoptosis and phosphorylation at S164/T165. However, similar Dox-induced RIPK3 expression in HeLa cells did not promote S164/T165 phosphorylation or apoptosis. The authors attributed this differential response to the lower expression of the RIPK3 chaperone HSP90/Cdc37 in MCF7 and KGN cells. The phosphomimetic mutant S164D/T165E RIPK3 was defective for classical TZB-induced necroptosis but induced apoptosis in HeLa cells despite high HSP90/CDC37 expression. By contrast, S164A/T165A-RIPK3 failed to cause apoptosis but restored TBZ-induced necroptosis in MCF7 and KGN cells. The authors propose that this process is particularly important in luteal granulosa cells and provide evidence suggesting that RIPK3 phosphorylation on S164/T165 occurs in the ovaries of older mice. This seems counterintuitive given that corpus luteum involution occurs as part of the ovulation cycle and should therefore be especially relevant in young, sexually mature mice.The authors are commended for the amount of work they have put in this study, going from the identification of this phosphorylation site to in vitro and in vivo functional studies including the generation of knock-in mice to address its function. The authors have successfully identified a protein phosphorylation that controls the form of cell death mediated by RIPK3. Key questions remain as to how this modification prevents RIPK3 from promoting necroptosis. There are some aspects of the study that could be strengthened, particularly regarding the mechanism of regulation and the physiological significance of this phosphorylation event.1. Using cells from the knock-in mice expressing the mutated proteins at endogenous levels would be a more appropriate experimental system to assess the mechanistic underpinnings such as the interaction with HSP90/CDC37. For example, in figure 5G, cell death can be assessed in BMDMs from both the phosphomimetic and the alanine mutants in a more comprehensive manner. Do BMDMs or other primary cells (MEFs, lymphocytes, etc) from the phosphomimetic mutant knock-in mice undergo spontaneous apoptosis in the absence of zVAD, and if not why?

We are fully aware the usual limitation of using ectopically expressed protein in studying cellular biology and appreciate the reviewers’ request to use the endogenous protein, including the mutant proteins from the knockin mice, to verify the findings. That is the precise reason we painstakingly searched various mouse tissues of different age using our phosphoSerine165/theronine166 specific antibody for the sign of endogenous activation of this pathway. So far, the luteal granulosa are the only cell type we found that naturally activates this pathway when mice reach to a certain age. We were able to recapitulate the activation event with primary granulosa cells cultured in vitro and in ovaries from hyper-ovulated mice with an addition of a prostaglandin_2α_ analog (Figure 7A-7C). Similar experiments are not possible with BMDM or fibroblasts. We also cordially disagree that further study of interactions between phospho-mimic and phosph-resistant mutant RIPK3 and HSP90/CDC37 would yield additional mechanistic insight in how RIPK3 adapts apoptotic and necroptotic functions. With this said, we did appreciate this comment and added a sentence in the text and in the method section stating that we do need to add caspase inhibitor when culture these BMDMs from the phosphor-mimic knock-in mice.

2. The authors implied that the kinase activity of RIPK3 is critical in this switch to apoptosis. However, the phenotypes of mice that lack RIPK3 kinase activity do not match that of the mice that harbor mutations that mimic this phosphorylation. Do RIPK3 deficient or S165a/T166A mutant mice show expected reproductive defects that may be attributed to the lack of the proposed RIPK3-mediated apoptosis program in luteal granulosa cells?

We have stated that the young RIPK3 deficient (KO) mice do not have fertility defect and therefore draw the conclusion that this pathway does not involve in regression of the corpus luteum during normal mouse reproductive cycle. This pathway seems to be specific for age-related, prostaglandin_2α_-induced apoptosis in luteal granulosa cells. We thus proposed in the Discussion that the effect of this pathway will only show up in aged animals. As the S165A/T166A mice are still young, and considering this manuscript is already data packed, we would like to publish the paper now and report the possible reproductive impact of the RIPK3 KO and KI mice when these mice grow older. We cannot anticipate such results in reasonable time frame because we do not know when the potential phenotype will become obvious. To clearly stating the fact that this pathway is only working in prostaglandin-induced luteal regression, not during normal reproductive cycle, we added “Prostaglandin_2α_-induced corpus luteal regression” in the title of the revised manuscript.

3. The heterozygous expression of the phosphomimetic mutants does not cause any pathology in vivo. The authors speculate that a threshold of expression is required for activation of this mutant, however, an alternative explanation could be that the presence of the wild-type protein prevents its activation, e.g. by trans-autophosphorylation on S227. Introducing a RIPK3 null allele to generate heterozygous RIPK3S165D/T166E mice that do not express wild-type RIPK3 could in principle help resolve this question, as in that case the phosphomimetic mutant will be expressed at the same level but in the absence of the wild type protein. We hope the authors can discuss this further.

We cannot exclude the reviewers’ explanation that the presence of the wild-type protein prevents its activation, e.g. by trans-autophosphorylation on S227 and agree that introducing a RIPK3 null allele to generate heterozygous RIPK3S165D/T166E mice that do not express wild-type RIPK3 will in principle help resolve this question. Given the time frame of generating such data and the fact that the outcome should not change the main conclusion of this manuscript, we chose to discuss these two possibilities in the text in our revised manuscript at current stage.

4. The in vivo data in the knock-in mouse models clearly show that phosphomimetic mutations (RIPK3S165D/T166E) on RIPK3 cause severe pathology in multiple organs associated with increased numbers of dying cells. However, definitive evidence that the pathology is induced by apoptosis is not provided. The authors should incorporate a statement that clarifies this or provide evidence implicating an apoptotic pathway. For instance, rescue experiments by crossing to caspase-8 knockout mice would be one mechanism to establish that pathology is indeed induced by apoptosis.

We did show that the defective tissues in *Ripk3^D161N/D161N^* mice had strong active caspase-3 signals that were absent in wild type mice, thus indicated that the pathology is caused by aberrant apoptosis induced by the phosphor-mimic RIPK3. Given that similar apoptosis phenotype could be rescued by cross with caspase-8 KO mice as previously report (Newton, K. et al. Science, 2014), and the time frame to perform such an experiment, we would like to do the proposed experiment for a future follow up publication but not in this manuscript.

5. The histological data and particularly the cleaved caspase-3 immunostainings are very hard to evaluate and could be improved. A question that deserves some discussion in the paper is also why the S165D/T166E mice don't die during embryonic life like the D161N mutants.

We believe that the immunostaining of cleaved caspase-3 signal is clear and the data may need to be seen under bigger amplification. Now with the high resolution electronic figures loaded, it can be easily done by the readers. As for why the S165D/T166E mice do not die during embryonic life like the D161N mutants, we believe that the readers should draw their own conclusion since the two forms of the mutant protein are not identical.

6. Information on the reproducibility of the experiments, including technical versus biological replicates and how many times each experiment was repeated, is not included in many figures. In addition, for all in vitro experiments in transfected and CRISPR knockout clones, it would help to include information on the number of independent clones tested. Statistical analysis needs to be included in all figures.

All experiments were repeated at least twice, with similar results and this information was provided in **Statistical analysis section.** We have provided more detail information in the figure legend in our revised manuscript.

7. RIPK3 immunoblots often show 2 or three bands, which sometimes seem to correspond to phosphorylated versions and other times FLAG-tagged versus wild type. It would be helpful to label these accordingly in the figures to make the data easier for the readers to assess.

We agree with the reviewers’ comments and label these accordingly in the figures.

[Editors' note: further revisions were suggested prior to acceptance, as described below.]

Essential revisions:1. Since the S165/T166 phosphorylation mutant does not fully phenocopy that of the RIPK3 kinase-inactive mutants, the possibility that the underlying mechanisms by which these mutants promote apoptosis (i.e. different phenotypes of the respective mutant mice) is not ruled out. The authors should clarify this further in the text.

We have updated subsection "The level of RIPK3 chaperone Hsp90/CDC37 determines the cellular necroptotic or apoptotic function of RIPK3" to clarify this further.